# Agent Reviewers: Domain-specific Multimodal Agents with Shared Memory for Paper Review

**Kai Lu** [* 1 2]  **Shixiong Xu** [* 1 2]  **Jinqiu Li** [* 1 2]  **Kun Ding** [1]  **Gaofeng Meng** [1 2 3]

## Abstract

Feedback from peer review is essential to improve the quality of scientific articles. However, at present, many manuscripts do not receive sufficient external feedback for refinement before or during submission. Therefore, a system capable of providing detailed and professional feedback is crucial for enhancing research efficiency. In this paper, we have compiled the largest dataset of paper reviews to date by collecting historical open-access papers and their corresponding review comments and standardizing them using LLM. We then developed a multi-agent system that mimics real human review processes, based on LLMs. This system, named Agent Reviewers, includes the innovative introduction of multimodal reviewers to provide feedback on the visual elements of papers. Additionally, a shared memory pool that stores historical papers' metadata is preserved, which supplies reviewer agents with background knowledge from different fields. Our system is evaluated using ICLR 2024 papers and achieves superior performance compared to existing AI-based review systems. Comprehensive ablation studies further demonstrate the effectiveness of each module and agent in this system. Our code and data are available at https://github.com/AReviewers/AgentReviewers.

## 1. Introduction

In academia, peer review is essential for validating research findings and improving manuscript quality. In recent years,

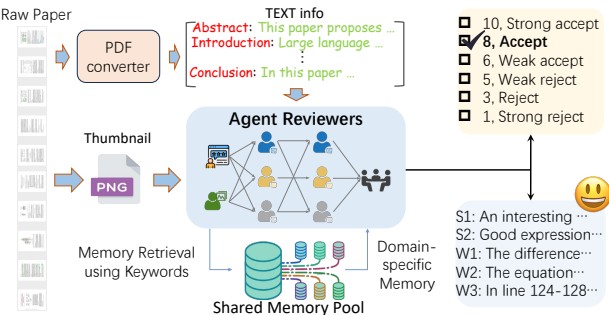

*Figure 1.* Agent Reviewers: a paper review system comprises domain-specific multimodal agents with shared memory. It leverages both paper text and thumbnails for evaluation, retrieving domain-specific memory from a shared memory pool to deliver decisions, scores, and insightful comments.

the growing number of researchers has led to an explosive rise in annual paper submissions. For example, in artificial intelligence, ICLR submissions surged from 490 in 2017 to 11,672 in 2025. However, the quantity and quality of peer reviews have failed to keep pace, leaving many young researchers, especially students, without the high-quality feedback needed to refine their work (Stelmakh et al., 2021; Zhang et al., 2022; Fox et al., 2023). As a result, many newcomers find themselves caught in an inefficient arms race, prioritizing submission volume over research quality.

Fortunately, the emergence of Large Language Models (LLMs) offers a promising solution to this prevalent issue. Many researchers have explored the use of LLMs to provide feedback on academic papers. Existing researches typically formulate automated peer review as a Supervised Fine-Tuning (SFT) task (Gao et al., 2024) or a zero-shot task based on LLMs (Yuan et al., 2022; Liu & Shah, 2023; Liang et al., 2024). The former relies on the collection and cleaning of real review data, while the latter depends on the adjustment of model prompts.

However, review comments generated by these methods often suffer from severe homogenization due to multiple factors. First, the same paper may receive diverse opinions from different reviewers, but a model fine-tuned on such data risks collapsing into uniform review comments. Sec-

---

[*]Equal contribution [1]State Key Laboratory of Multimodal Artificial Intelligence Systems, Institute of Automation, Chinese Academy of Sciences, Beijing, China [2]School of Artificial Intelligence, University of Chinese Academy of Sciences, Beijing, China [3]The Centre for Artificial Intelligence and Robotics, HK Institute of Science & Innovation, Chinese Academy of Sciences. Correspondence to: Gaofeng Meng <gfmeng@nlpr.ia.ac.cn>.

*Proceedings of the 42nd International Conference on Machine Learning*, Vancouver, Canada. PMLR 267, 2025. Copyright 2025 by the author(s).

ond, current approaches heavily rely on the model's internal domain knowledge, which is often insufficient for generating constructive and distinctive feedback. Lastly, these methods typically assess papers based solely on text, whereas real peer review also considers figures, layout, and other visual elements. The absence of visual modalities further reduces the diversity of review comments (Von Bearnensquash, 2010; Huang, 2018). As a result, AI-generated homogenized comments could not provide insightful feedback before submission. Moreover, their increasing presence in academic journal and conference reviews significantly diminishes the submission experience for authors.

Inspired by the aforementioned analysis, we propose Agent Reviewers for automated paper review, as shown in Figure 1. This framework fundamentally emulates real-world peer reviewers' expertise acquisition patterns: By constructing multiple reviewers with distinct roles and modalities, it mirrors how actual reviewers leverage domain-specific knowledge related to paper keywords, ultimately generating high-quality, insightful evaluations. The system architecture embodies three core innovations: Multi-agent Interaction (MI), Shared Memory Pool (SMP) designed to mimic reviewers' background knowledge repository, and Multimodal Agent (MA). Specifically, the SMP and domain-specific agent design originate from simulating human reviewers' practice: the Meta-reviewer agent extracts paper keywords to initialize specialized reviewers through SMP, which systematically organizes domain knowledge akin to real reviewers' interests. Concurrently, the MA processes PDF/PNG files to critique visual elements (Von Bearnensquash, 2010; Huang, 2018). These domain-specific agents then conduct multi-perspective assessments, generating preliminary reviews. After iterative discussion and comments refinement, a chair reviewer agent summarizes the entire review process to provide the final review comments and ratings. These review comments will serve as meta-information for the paper and update the shared memory pool.

To facilitate this study, a dataset with standard reviews is proposed in Section 4, termed Reviews-STD. It is derived from Reviewer2 (Gao et al., 2024) and SEA (Yu et al., 2024), including papers from the International Conference on Learning Representations (ICLR) spanning 2017 to 2024 and the Conference on Neural Information Processing Systems (NeurIPS) covering 2016 to 2024. To enhance consistency and accessibility, we apply the ERNIE-Speed-128K model to transform disparate reviews into a unified format, which consists of two lists (strengths and weaknesses) and the decision. For the test set, we use 100/300 evenly randomly selected papers from ICLR 2024.

Once the shared memory pool is initialized using Reviewer-STD, we conducte evaluations on the test set, where Agent Reviewers demonstrates superior performance compared

with state-of-the-art (SOTA) methods, in terms of decision prediction and review quality scoring. Finally, extensive ablation experiments are conducted to validate the effectiveness of each module within the system.

Our contributions are summarized as follows: (1) We constructed a multi-agent review system, named Agent Reviewers, which consists of a shared memory pool for domain-specific reviewers initialization and a multimodal agent to enhance the diversity of the generated comments. (2) We proposed a meticulously standardized large-scale paper review dataset, Reviewer-STD, along with a benchmark from ICLR 2024. (3) Agent Reviewers significantly outperforms existing methods in terms of paper acceptance prediction and the quality of paper review comments.

## 2. Related work

### 2.1. AI for Review

Recent advances in AI, particularly LLMs, have created new opportunities for scientific research (Baek et al., 2024; Wang et al., 2024b; Huang et al., 2024; Wang et al., 2024c; D'Arcy et al., 2024; Wei et al., 2023). Peer review, a critical research component, has gained attention, with several studies leveraging LLMs for this task. (Yuan et al., 2022) explores using NLP models for initial peer reviews and discusses evaluation metrics. (Liu & Shah, 2023; Liang et al., 2024) assess the reliability of LLM-generated reviews through large-scale analyses with carefully designed prompts. The AI Scientist framework (Lu et al., 2024) introduces a fully automated scientific pipeline, covering idea generation, paper writing, and a simulated review process with self-reflection and aggregation. (Du et al., 2024) compares human-written and LLM-generated reviews at the sentence level, analyzing LLMs' potential as reviewers and meta-reviewers. (Zhou et al., 2024) presents the RR-MCQ dataset for review-revision tasks, evaluating GPT-3.5 and GPT-4 in score prediction and review generation. REVIEWER2 (Gao et al., 2024) proposes a two-stage review generation framework using prompt guidance and fine-tunes LongLoRA on a large peer review dataset. SEA (Yu et al., 2024) standardizes reviews into a unified format, fine-tuning Mistral-7B and employing a self-correcting strategy to enhance quality. (Weng et al., 2024) introduces an autonomous research and review framework, where CycleResearcher conducts research while CycleReviewer simulates peer review, providing iterative feedback via reinforcement learning. Unlike prior works focusing on text, (Von Bearnensquash, 2010; Huang, 2018) employ AdaBoost and deep convolutional networks to predict paper acceptance or rejection based on visual features, underscoring the importance of visual information.

Collectively, these efforts highlight the potential of AI, including LLM-based and vision-based approaches, to assist

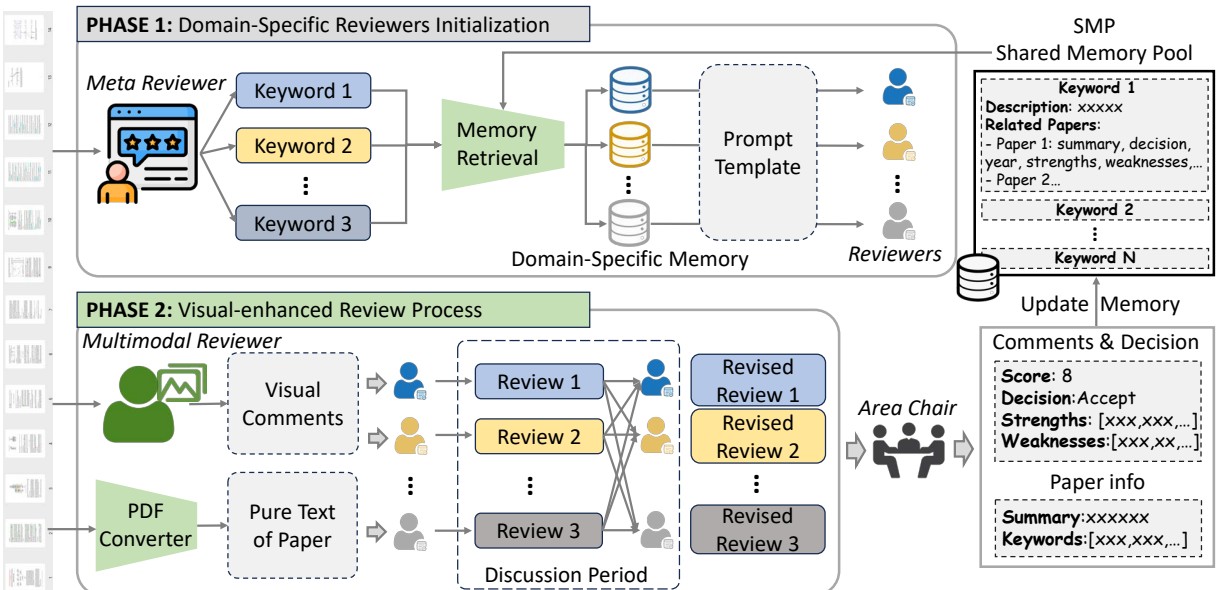

*Figure 2.* Overview of the Agent Reviewers system. In *PHASE 1*, the meta reviewer extracts keywords from the paper text and retrieves memory from the SMP to initialize domain-specific reviewers. In *PHASE 2*, a multimodal reviewer provides visual comments, and domain-specific reviewers integrate these with the paper text for initial review and discussion to refine their assessments. Finally, the AC consolidates all revised reviews to produce the final comments and decision, updating the SMP with the paper information and review.

and enhance the traditional scientific review process. However, prior research often relies on single-agent frameworks and treats textual and visual data in isolation. In our study, we advance this by integrating multi-agent collaboration and multimodal capabilities to leverage both textual and visual information effectively.

### 2.2. Multi-agent System

In recent years, multi-agent systems based on large language models (LLMs) have demonstrated exceptional capabilities and promising applications. By leveraging specialized agents with distinct functions to collaborate and interact, multi-agent systems have found broad applications in fields such as software engineering (Huang et al., 2023; Qian et al., 2023; Jin et al., 2024a), society simulation (Park et al., 2023; Li et al., 2024), and embodied intelligence (Mandi et al., 2024; Chang et al., 2024). Currently, multi-agent systems can be roughly categorized into two types from an application perspective: collaborative task execution and scenario simulation. Collaborative task execution aims to solve complex problems through cooperation among specialized agents. For example, (Khan et al., 2024) uses two expert models to engage in a debate, with a non-expert model acting as a judge to select a side, improving the accuracy of the non-expert model on reading comprehension tasks. (Huang et al., 2023) designed a multi-agent code generation framework consisting of a programmer agent, a test designer agent, and a test executor agent, achieving efficient and low-cost code generation. On the other hand,

scenario simulation focuses on using multi-agent systems to model real-world scenarios. For example, (Park et al., 2023) created a small town with 25 autonomous agents to simulate human behavior, while (Chang et al., 2024) developed a simulated hospital system with patient, nurse, and doctor agents, where the treatment performance of the doctor agent gradually improves as the simulation evolves.

Recently, multi-agent systems have been increasingly applied to scientific research, with studies such as (Ghafarollahi & Buehler, 2024) integrating them with ontological knowledge graphs to support materials development. Another example is (Jin et al., 2024b), which developed a multi-agent peer review system to simulate and study psychological phenomena in peer review. In contrast, our work focuses on enhancing the design of multi-agent systems by introducing a shared memory pool and multimodal capabilities, to leverage multi-agent collaboration to generate high-quality review feedback.

## 3. Agent Reviewers

### 3.1. System Overview

We overview our proposed system in Figure 2. Agent Reviewers comprise four categories of agent roles and two main phases: Domain-Specific Reviewers Initialization and Visual-enhanced Review Process.

**Domain-Specific Reviewers Initialization.** During the Domain-Specific Reviewers Initialization phase, the meta re-

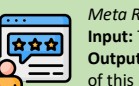 *Meta Reviewer*
**Input:** Text of paper
**Output:** Several keywords of this paper

**Responsibility:** Extracting a carefully selected set of keywords from the paper's content, which should over the core topics and expertise required to evaluate the submitted paper.

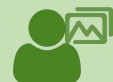 *Multimodal Reviewer*
**Input:** Thumbnail PNG of the paper
**Output:** Comments about the layout.

**Responsibility:** Evaluates the paper's visual appeal, formatting, and quality of visual elements such as charts and images, providing feedback to assist other agents in the review process.

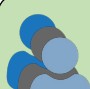 *Domain-specific Reviewers*
**Input:** Text of paper, comments from multi-modal reviewer, reviews from other reviewers
**Output:** Comments, score, and decision
**Responsibility:** Generating the {Attention/NLP/Optimization}-related reviews, discussing with other reviewers, modifying the reviews accordingly

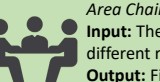 *Area Chair Reviewer*
**Input:** The reviews from different reviewers
**Output:** Final reviews, score, and decision
**Responsibility:** Synthesizing final comments from domain-specific reviewers to in two mode: single-paper and batch review.

*Figure 3.* Key roles in Agent Reviewers system.

viewer agent extracts $k$ keywords from the manuscript. For each keyword, historical paper information (e.g., summaries, decisions) is retrieved from a shared memory pool and incorporated into a prompt template to initialize a domain-specific reviewer with expertise in the corresponding field. This process generates $k$ specialized reviewers.

**Visual-enhanced Review Process.** In this phase, the multi-modal reviewer first provides comments on the visual content of the manuscript based on the thumbnail of the paper. A PDF converter is utilized to obtain a plain-text version of the manuscript. The visual comments and the plain text manuscript are then fed into the domain-specific reviewers to elicit a variety of review opinions. This is followed by the discussion phase, where each reviewer reads and considers the feedback from others to revise their own reviews. The final revised reviews are submitted to the Area Chair (AC) to compile the final comments on the manuscript and to gather information for updating the shared memory pool.

This structured system ensures that the paper undergoes a comprehensive evaluation, incorporating both textual and visual elements, leading to well-rounded and informed review outcomes. The introductions of the different roles and their interaction, together with the design of the shared memory pool are detailed below.

### 3.2. Agent Roles

**Meta reviewer agent.** The meta reviewer agent is responsible for extracting a set of keywords from the paper's content, which are then used to initialize a group of domain-specific reviewer agents. These keywords are thoughtfully chosen to cover the core topics and expertise required to evaluate the submitted paper, ensuring that each domain-specific reviewer agent possesses the necessary domain knowledge to provide thorough and insightful feedback.

**Mulit-modal reviewer agent.** Existing LLM review methods typically convert papers into plain text, ignoring crucial visual information such as formatting and figures (Huang, 2018). Recent multimodal models can leverage visual information, but high-resolution document images require extensive token usage, making visual token integration computationally expensive. To address this, we developed a

specialized multimodal reviewer agent to extract layout and other visual cues relevant to the review. Specifically, we represent the document's main text as a 3×3 thumbnail, serving as input to the multimodal reviewer. We find that this approach effectively balances minimal visual token usage with accurate layout assessment and content understanding. The agent evaluates visual appeal, formatting, and the quality of visual elements such as charts and images, providing feedback to support other agents in the review process.

**Domain specific reviewer agent.** Peer review significantly benefits from diverse expertise, as reviewers typically have varied backgrounds. To mimic this diversity, we employ multiple domain-specific reviewer agents for each paper. These agents are specialized in distinct subfields, assigned by the meta reviewer agent based on keywords extracted from the paper. Each agent is initialized with domain-specific memory from a shared memory pool to assess the paper's strengths, weaknesses, and overall quality, providing scores and acceptance recommendations.

**Chair reviewer agent.** The chair reviewer agent combines paper information and domain-specific agents' reviews to deliver the final score, decision and comments, and it also updates the relevant keyword entries in the shared memory pool, enabling the dynamic expansion of the pool.

### 3.3. Agent Interactions

Inspired by the peer review process, we have subdivided the two-phase review workflow into the following 5 interaction periods. The domain-specific reviewers initialization phase corresponds to period **I**, while the visual-enhanced review process encompasses periods **II** through **V**.

**I. Domain Specific Reviewer Agent Initialization.** The meta reviewer agent extracts $k$ keywords from the paper text to initialize $k$ domain-specific reviewer agents, each specializing in a distinct field, using relevant knowledge retrieved from the shared memory pool.

**II. Visual Information Extraction.** The multimodal reviewer agent evaluates the paper's visual aspects (e.g., layout, charts, and figures) based on its thumbnail. This feedback serves as supplementary information for the subse-

quent review phases.

**III. Initial Review.** Each domain-specific reviewer agent generates an initial review by integrating their area-specific knowledge with the paper's text and visual information, assessing both strengths and weaknesses.

**IV. Discussion and Revision.** Initial reviews are shared with all agents, allowing them to revise their feedback based on peer insights. This process refines the initial reviews.

**V. Final Review and Memory Update.** The chair reviewer agent synthesizes paper information and domain-specific reviews to generate the final comments and decision, while also updating relevant keyword entries in the SMP.

### 3.4. Shared Memory Pool

To equip the domain-specific reviewer agent with specialized knowledge, we have designed a precise RAG-based Shared Memory Pool (SMP).For each keyword, the pool stores information on related papers, including their summaries, strengths and weaknesses, submission years, decisions, and other pertinent details. When the meta reviewer agent assigns a keyword to a domain-specific agent, relevant paper knowledge can be retrieved from the pool and provided to the domain-specific agent.

**Initialization.** Before evaluating Agent Reviewers on the ICLR 2024 test set, we initialize the shared memory pool with papers from Reviews-STD before ICLR 2024 to ensure comprehensive knowledge coverage. Using the free GLM-4-Flash API, we generate a summary and several keywords for each paper, along with a description for each keyword. To integrate the keywords, we design a two-stage aggregation pipeline inspired by Instag (Lu et al., 2023). (1) Rule-based Aggregation. Keywords are normalized (lowercased and cleaned of special characters) and processed with NLTK's stemming to merge synonyms like "conditional generation" and "conditional generator." (2) Semantics-based Aggregation. Keywords are encoded into semantic embeddings, and DBSCAN is used to cluster similar keywords. We found that existing embedding models struggle with AI-specific terms (e.g., overemphasizing surface similarity, linking "recurrent neural network" and "graph neural network" while separating "diffusion model" and "VAE"), so we use keyword descriptions for embedding, encoded by `all-mpnet-base-v2`[1] model. Finally, GLM-4-Flash creates a new keyword and it's corresponding description for each cluster. The semantics-based aggregation can be iterated for results at different granularities.

**Retrieval.** During the review process, the meta-reviewer extracts keywords from the paper under review and retrieves

---

[1] https://huggingface.co/sentence-transformers/all-mpnet-base-v2

*Table 1.* Dataset Comparison

| | # papers | # reviews | standardized review format |
|---|---|---|---|
| PeerRead (Kang et al., 2018) | 3,006 | 10,770 | ✗ |
| ASAP-Review (Yuan et al., 2022) | 8,877 | 28,119 | ✗ |
| MReD (Shen et al., 2022) | 7,894 | 30,764 | ✗ |
| NLPeer (Dycke et al., 2023) | 5,672 | 11,515 | ✗ |
| Reviewer2 (Gao et al., 2024) | 27,805 | 99,727 | ✗ |
| SEA (Yu et al., 2024) | 12,296 | 47,602 | ✓ |
| Reviews-STD | 38,312 | 144,027 | ✓ |

memory about related papers from the Shared Memory Pool (SMP) to initialize the domain-specific reviewer. We design a two-stage process for fine-grained retrieval: (1) Keywords Retrieval. The meta-reviewer uses keyword description embeddings to search the SMP and retrieves the top-$k_1$ most similar keywords and their associated papers as candidates. (2) Paper Reranking. Candidate papers are reranked based on the similarity between their summary embeddings and that of the paper under review, with the top-$k_2$ most similar papers selected as supplementary knowledge for the domain-specific reviewer. After the review, the reviewed paper information is updated to the Shared Memory Pool.

## 4. Reviews-STD

We incorporate selected data from the Reviewer2 (Gao et al., 2024) and SEA (Yu et al., 2024) datasets, which include papers from the International Conference on Learning Representations (ICLR) (2017–2024) and the Conference on Neural Information Processing Systems (NeurIPS) (2016–2024). For each paper, we retain metadata such as the full content, reviews, meta-reviews, and final decisions. Notably, these datasets are sufficiently dated to have potentially been included in the training datasets of foundational LLMs. Although prior research (Lu et al., 2024) suggests LLMs have not memorized this data, the lack of transparency in publicly available training datasets prevents confirming this. To ensure the robustness of our results, we also collect ICLR 2024 submissions from OpenReview[2], retaining relevant metadata for each paper. Additionally, we employ MinerU (Wang et al., 2024a) to convert PDFs into JSON files.

**Standardization.** Each paper in the dataset is typically accompanied by 3–5 reviews. Given the variability in review formats and evaluation criteria across conferences and years, directly using this data without preprocessing could

---

[2] https://openreview.net

introduce biases into evaluation outcomes. To enhance consistency and accessibility, we apply the ERNIE-Speed-128K model to unify diverse reviews into a standardized format. This standardized format comprises two lists: strengths and weaknesses. Strengths highlight features supporting a paper's acceptance, while weaknesses identify points that could lead to rejection. Each list item is accompanied by a distinct comment to avoid redundancy. As shown in Table 1, our dataset provides a more consistent format and contains a larger, more comprehensive dataset compared to other publicly available datasets. The prompts used for review summarization are detailed in Appendix D, and the statistics of the proposed Reviews-STD are in Appendix A.

## 5. Experiments

### 5.1. Experimental Setup

We use GPT-4o-mini as the core LLM unless otherwise specified, due to its balanced knowledge base, multimodal capabilities, and cost-effectiveness. By default, we use "title + abstract + introduction" as the paper content, employ 3 domain-specific reviewer agents, and keep the shared memory pool frozen during test time.

**Evaluation Dataset.** We conducted our evaluation on ICLR 2024. The dataset offers a relatively balanced distribution between accepted and rejected papers, and its release date postdates the knowledge cutoff of the core LLM, eliminating the risk of data leakage. We randomly selected 300 papers from ICLR 2024 for the main experiments, and 100 papers for ablation studies, balancing evaluation sufficiency with experimental cost. To ensure consistency, accepted and rejected papers were sampled in accordance with ICLR 2024's overall acceptance rate of approximately 40%.

**Shared Memory Pool.** We initialized the shared memory pool with all 28372 papers from ICLR 2017-2023 and NeurIPS 2016-2023 in Reviewer-STD, ensuring the SMP provides domain-specific agents with comprehensive knowledge. We used the free GLM-4-Flash API, extracting 3 keywords per paper, resulting in a total of 43954 keywords after 3 rounds of merging(DBSCAN: $\epsilon = 0.25$, min_samples = 2), with an average of 1.78 papers per keyword. During retrieval, the parameters top-$k_1$ for keywords and top-$k_2$ for papers as final memory are both set to 5.

**Compared Methods.** We compare our method with AI-Scientist (Lu et al., 2024), AgentReview (Jin et al., 2024b), and LLM Review (Liang et al., 2024), using the same LLM (GPT-4o-mini) for all methods to ensure fair comparison. Other settings follow their respective defaults (see Appendix B.2 for details). Neither LLM Review nor AgentReview explicitly lists their strengths and weaknesses. To enable a standardized comparison with our method, we use GPT-4o-mini to extract the strengths and weaknesses of the

reviews, which are then used as the strengths and weaknesses for these two methods.

### 5.2. Evaluation Metrics

We evaluate the performance of our method in two key aspects: the quality of generated strengths and weaknesses and the accuracy of decisions.

**Strengths-Weaknesses Analysis.** Strengths-weaknesses quality analysis focuses on how well the strengths and weaknesses identified by the LLM align with those of the human reviewers. Evaluating the quality of LLM-generated feedback at the review level is particularly challenging, as a strong LLM review may represent a synthesis of perspectives from multiple reviewers, yet exhibit low similarity to any single one of them. To address this, we focus on fine-grained evaluation by extracting the strengths and weaknesses from the ground truth review comments in the dataset. This allows us to break down the review-level assessment into a more detailed analysis at the strengths and weaknesses level. Specifically, we use the `all-mpnet-base-v2`[3] model to encode the strengths and weaknesses into embeddings, and then calculate the cosine similarity. We evaluate the extent to which LLM-generated strengths and weaknesses align with those identified by human reviewers, ensuring high-quality feedback while minimizing irrelevant or inaccurate additions. To achieve this, we design and employ F1-score and Jaccard to balance coverage and accuracy, supplemented by Recall and MaxSim to further assess coverage. Detailed metric definitions are provided in Appendix B.1.

**Decisions Analysis.** We approach this as a binary classification problem. However, the inherent class imbalance due to the low acceptance rate may result in biased evaluations. To address this, we employ robust metrics designed for imbalanced datasets, including F1-score, Matthews Correlation Coefficient (MCC), Balanced Accuracy (Bal. Acc), and G-mean, ensuring a more reliable assessment. Detailed definitions are provided in Appendix B.1.

### 5.3. Main Results

Table 2 compares the performance of Agent Reviewers with existing methods across multiple metrics. More detailed comparison results, including additional evaluation metrics, are provided in Appendix B.3.

**Strengths-Weaknesses Analysis.** Using GPT-4o-mini, Agent Reviewers outperforms the current best approach across multiple metrics, achieving a 10.5% improvement in total Jaccard and an 8.5% increase in F1-score, demonstrating a stronger alignment between the strengths and weaknesses generated by our method and those identified

---

[3] https://huggingface.co/sentence-transformers/all-mpnet-base-v2

*Table 2.* Main results for strengths-weaknesses and decisions analysis. Str. and Wk. denote strengths and weaknesses, respectively. F1-score and Jaccard in strengths-weaknesses analysis are calculated with a similarity threshold of 0.5. **Bold** indicates the best performance, underline the second-best. AgentReview(Top-$k$) accepts the top-$k$ ranked papers per batch of 10 papers. All methods use GPT-4o-mini.

| Method | Strengths-Weaknesses Analysis | | | | | | Decisions Analysis | | | | | Cost($)↓ |
| | F1-score↑ | | | Jaccard↑ | | | Decision | F1-score↑ | MCC↑ | Bal. Acc↑ | G-mean↑ | /100 Papers |
| | Total | Str. | Wk. | Total | Str. | Wk. | Offered? | | | | | |
|---|---|---|---|---|---|---|---|---|---|---|---|---|
| AgentReview(Top-3) (Jin et al., 2024b) | 0.340 | 0.442 | 0.245 | 0.215 | 0.252 | 0.194 | ✓ | 0.410 | 0.104 | 0.549 | 0.515 | 7.53 |
| AgentReview(Top-4) (Jin et al., 2024b) | 0.340 | 0.442 | 0.245 | 0.215 | 0.252 | 0.194 | ✓ | 0.417 | 0.028 | 0.514 | 0.505 | 7.53 |
| AI_Scientist (Lu et al., 2024) | 0.426 | 0.523 | 0.345 | 0.285 | 0.313 | 0.272 | ✓ | 0.049 | 0.123 | 0.513 | 0.158 | 3.59 |
| LLM Review (Liang et al., 2024) | 0.420 | 0.560 | 0.328 | 0.275 | 0.243 | **0.336** | ✗ | - | - | - | - | **0.13** |
| Agent Reviewers | **0.462** | **0.577** | **0.357** | **0.315** | **0.333** | 0.310 | ✓ | **0.566** | **0.220** | **0.613** | **0.612** | 0.65 |

*Table 3.* Ablation study for strengths-weaknesses analysis. Each metric is a comprehensive measure of both strengths and weaknesses, corresponding to the "Total" column in Table 2.

| Method | Recall↑ | F1-score↑ | MaxSim↑ | Jaccard↑ |
|---|---|---|---|---|
| Baseline | 0.347 | 0.416 | 0.442 | 0.270 |
| w/o Multi-Agent | 0.335 | 0.404 | 0.435 | 0.260 |
| w/o SMP | 0.364 | 0.412 | 0.456 | 0.275 |
| w/o Multimodality | 0.394 | 0.436 | 0.463 | 0.296 |
| w/o Discussion | 0.389 | 0.424 | 0.472 | 0.282 |
| Agent Reviewers | **0.409** | **0.453** | **0.476** | **0.307** |

*Table 4.* Ablation study for decisions analysis.

| Method | Acc↑ | F1-score↑ | MCC↑ | Bal. Acc↑ | G-mean↑ |
|---|---|---|---|---|---|
| Baseline | 0.66 | 0.469 | 0.219 | 0.609 | 0.593 |
| w/o Multi-Agent | 0.63 | 0.413 | 0.143 | 0.571 | 0.547 |
| w/o SMP | 0.66 | 0.292 | 0.110 | 0.543 | 0.436 |
| w/o Multimodality | 0.65 | 0.568 | 0.314 | 0.668 | 0.666 |
| w/o Discussion | 0.51 | 0.424 | 0.045 | 0.524 | 0.522 |
| Agent Reviewers | **0.70** | **0.605** | **0.385** | **0.705** | **0.705** |

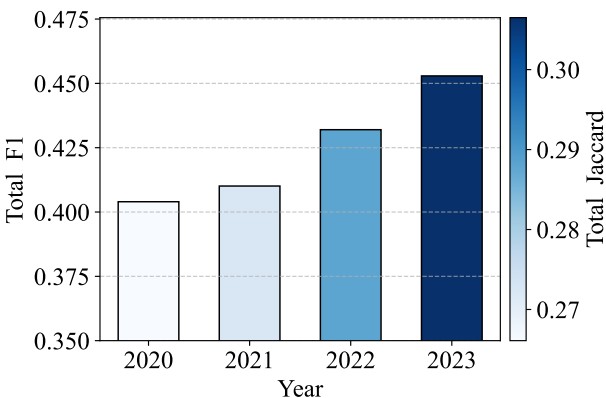

*Figure 4.* Impact of the cutoff year of shared memory pool(SMP) on strengths and weaknesses quality.

by human reviewers. Our Agent Reviewers strike a well-balanced trade-off between coverage and precision, delivering consistently strong performance across all metrics. Appendix B.7 provides a further analysis of strengths and weaknesses, discussing their distribution across categories as well as differences in alignment with human reviewers.

**Decisions Analysis.** Using GPT-4o-mini, our method ranks first across all four metrics, significantly outperforming existing approaches. F1-score, MCC, Bal. Acc, and G-mean achieve relative improvements of 35.7%, 78.9%, 11.7%, and 18.8% over the second-best method, respectively, demonstrating superior alignment with human reviewers' decision patterns. These substantial gains suggest that Agent Reviewers better capture human reviewers' preferences, leading to more consistent and reliable decisions.

**Cost Comparison.** Table 2 demonstrates that Agent Reviewers not only achieves strong performance, but also maintains low costs. It results from our efficient process design, which avoids excessive iterations and delivers excellent results even on cost-effective models like GPT-4o-mini.

### 5.4. Ablation Studies

We conducted ablation experiments on the key modules of Agent Reviewers to systematically assess their individual contributions to review quality and decision accuracy. Tables 3 and 4 present the results for strengths-weaknesses analysis and acceptance decision metrics, respectively. "Baseline" refers to a text-only single agent directly generating final reviews. "w/o Multimodality" excludes visual information, relying solely on textual data to review. "w/o Multi-Agent" removes multi-agent interactions, using a single multimodal reviewer to process both text and vision tokens, then directly generate the final comments and decision. "w/o SMP" eliminates the shared memory pool for domain-specific agents. "w/o Discussion" removes the discussion period, forcing domain-specific agents to submit initial review comments directly to the AC without refinement. Furthermore, we assess the generalizability of our method across different LLM APIs, as shown in Table 5, demonstrating its adaptability across varying LLMs. Appendix B.6 presents the evaluation results across different conferences and years.

**Effects of Multi-Agent Interaction.** As shown in Tables 3 and 4, the removal of multi-agent interaction results in

*Table 5.* Comparison between proprietary and open-source LLMs.[†]is text-only LLM, we use GPT-4o-mini for multimodal agent.

| LLM | | Strengths-Weaknesses Analysis | | | | | | Decisions Analysis | | | |
| --- | --- | --- | --- | --- | --- | --- | --- | --- | --- | --- | --- |
| | | F1-score↑ | | | Jaccard↑ | | | F1-score↑ | MCC↑ | Bal. Acc↑ | G-mean↑ |
| | | Total | Str. | Wk. | Total | Str. | Wk. | | | | |
| Proprietary | GPT-4o-mini | 0.453 | 0.558 | 0.361 | 0.307 | **0.333** | 0.296 | **0.605** | **0.385** | **0.705** | **0.705** |
| | Gemini-exp-1206 | 0.430 | 0.561 | 0.327 | 0.289 | 0.283 | 0.304 | 0.536 | 0.367 | 0.668 | 0.638 |
| Open-source | Deepseek-V3-1226[†] | **0.461** | **0.584** | 0.358 | **0.310** | 0.275 | **0.356** | 0.531 | 0.223 | 0.594 | 0.484 |
| | Deepseek-V3-0324[†] | 0.458 | 0.564 | 0.370 | 0.306 | 0.288 | 0.332 | 0.552 | 0.299 | 0.618 | 0.485 |
| | Qwen2.5-VL-72B-Instruct | 0.450 | 0.554 | 0.362 | 0.303 | 0.290 | 0.327 | 0.532 | 0.222 | 0.600 | 0.516 |
| | InternVL-2.5-78B | 0.450 | 0.540 | **0.373** | 0.301 | 0.303 | 0.309 | 0.526 | 0.210 | 0.586 | 0.470 |

*Table 6.* Decision Analysis Using AC aggregation and Majority Voting (M.V.). *Initial*: M.V. based on initial reviews before discussion; *Revised*: M.V. based on revised reviews after discussion.

| | Acc↑ | F1-score↑ | MCC↑ | Bal. Acc↑ | G-mean↑ |
| --- | --- | --- | --- | --- | --- |
| M.V. (Initial) | 0.53 | 0.484 | 0.136 | 0.572 | 0.560 |
| M.V. (Revised) | 0.57 | 0.517 | 0.206 | 0.609 | 0.599 |
| AC Aggregation | **0.70** | **0.605** | **0.385** | **0.705** | **0.705** |

*Table 7.* Ablation study on memory retrieval methods. *Keyword* refers to retrieval based on keywords, while *Random* refers to random sampling. Both methods utilize 5 papers as memory.

| Retrieval | Str. & Wk.↑ | | Decisions↑ | | | |
| --- | --- | --- | --- | --- | --- | --- |
| | F1-score | Jaccard | F1-score | MCC | Bal. Acc | G-mean |
| Keyword | **0.453** | **0.307** | **0.605** | **0.385** | **0.705** | **0.705** |
| Randam | 0.417 | 0.278 | 0.400 | 0.054 | 0.528 | 0.525 |

an average relative decline of 13% in strengths-weaknesses metrics and 29% in acceptance decision metrics compared to the full method, emphasizing its pivotal role in enhancing review quality and decision accuracy. Furthermore, eliminating the discussion period significantly degrades both strengths-weaknesses evaluation and acceptance decision consistency, underscoring the importance of collaborative discussions in refining reviews. To study the role of the AC reviewer in final decisions, Table 6 compares AC-aggregated decisions with majority voting among domain-specific reviewers.AC-considered decisions show higher consistency with human reviewers than majority voting.

**Effects of Shared Memory Pool.** As shown in Tables 3 and 4, removing the SMP results in an average 9% relative decrease in strengths-weaknesses metrics and a 38% relative decrease in acceptance decision metrics compared to our full method. To further investigate the role of SMP, Table 7 shows that retrieving the top-5 papers by keywords significantly outperforms randomly sampling 5 papers as memory, highlighting the importance of precise domain-specific knowledge retrieval. Figure 4 illustrates the impact of initializing SMP with papers from different cutoff years on the quality of strengths and weaknesses. Later cutoff years result in a more comprehensive and up-to-date SMP, enhancing the overall quality of review comments.

**Effects of Multimodality.** Tables 3 and 4 show that removing multimodality reduces both strengths-weaknesses review quality and acceptance decision accuracy, underscor-

ing the multimodal reviewer's importance. We attribute this to its ability to assess document formatting and evaluate images and tables, leading to more relevant strengths, weaknesses, and decisions.

**Ablation Study on Different LLMs.** Table 5 presents the performance of our method across various open-source and proprietary LLMs, including GPT-4o-mini, Gemini-exp-1206, Deepseek-V3, Qwen2.5-VL-72B-Instruct, and InternVL-2.5-78B. Our approach demonstrates strong generalization across different LLMs: Deepseek-V3-1226 achieves the highest quality in analyzing strengths and weaknesses, while GPT-4o-mini shows the closest alignment with human reviewers in decision-making. Appendix B.4 compares Agent Reviewers and the Single Agent baseline under different LLMs, while Appendix B.5 explores the use of different LLMs for domain-specific reviewer roles.

### 5.5. Case Study

Figure 5 presents selected weaknesses for *KAN: Kolmogorov–Arnold Networks*. By citing specific prior works (e.g., ICLR 2017, 2021, 2022) and identifying overlooked research, our system leverages the Shared Memory Pool (SMP) to produce contextually rich and well-informed critiques. Further details and additional case studies on LoRA and this paper are provided in Appendix C.

1. The paper does not sufficiently leverage or discuss existing research on learnable activation functions (e.g., ICLR 2017, ICLR 2021, ICLR 2022) or related concepts, limiting the context and perceived novelty of the proposed approach.
2. The paper does not leverage the findings from previous research papers about 'Kolmogorov-Arnold Networks (KANs)', such as 'ICLR_2023_3364' which proposed a novel theoretical architecture, or 'ICLR_2022_1438' which applied Koopman operator theory to neural sequential models…

*Figure 5.* Weaknesses Identified for *KAN: Kolmogorov–Arnold Networks* Using Gemini-Exp-1206.

## 6. Conclusion

We constructed a multi-agent review system called Agent Reviewers. It is built on LLMs to mimic the human peer review process by creating multiple agents with different roles. It includes a meta-reviewer agent for initializing reviewers, a multimodal reviewer agent for perceiving visual content in

papers, domain-specific agents responsible for paper review, and an area chair agent for summarizing review comments. To address the common issues of homogenization and unprofessional review comments in AI-based peer review, we designed a shared memory to initialize reviewers with different backgrounds and introduced a reviewer communication phase to revise review opinions. Additionally, we introduced a multimodal reviewer agent to provide qualitative comments on the visual elements of the papers. Meanwhile, we collected a large-scale and standardized dataset called Reviews-STD to facilitate our study and built a benchmark using 300 randomly selected ICLR 2024 papers to evaluate and compare different methods. Agent Reviewers demonstrated excellent performance across multiple metrics. This system can serve as an important tool to provide insightful feedback for researchers to enhance paper quality.

**Limitations.** While our system achieves promising results, it has some limitations. First, we focus only on papers from two major AI conferences, limiting the scope of the memory. This training-free method can generalize to other domains by expanding the memory accordingly. Second, the current memory design is relatively simple and could be enhanced with techniques from retrieval-augmented generation (RAG) or integrating online search. Finally, the system has not undergone any fine-tuning; theoretically, its performance could be further improved by fine-tuning specific agents, such as the Meta-reviewer or Area Chair reviewer. We hope that this work can gain the attention of the community and collaboratively address these limitations in the future.

## Impact Statement

Peer review is essential for improving the quality of scientific research, yet many papers, especially those from early-career researchers, lack sufficient feedback before or during submission. To address this, we introduce Agent Reviewers, a multi-agent LLM-based system that emulates real peer review dynamics. By incorporating a multimodal reviewer and a shared memory pool, our multi-agent system delivers detailed and insightful feedback, aiding researchers in refining their work.

However, AI-assisted peer review comes with ethical responsibilities. The integrity of peer review depends on fairness, expertise, and human judgment. While our system can enhance and support the review process, it is not designed to replace human reviewers, as LLMs can still produce biased, inconsistent, or overly generic feedback. Human oversight remains essential to ensure fairness and reliability. Additionally, peer review data is inherently sensitive, and any deployment of AI-driven review systems must carefully address data privacy and confidentiality concerns. We strongly advocate for the responsible use of AI in peer review and emphasize the need for continued discussion on ethical safeguards and best practices for its integration.

## Acknowledgements

This work was supported partially by the National Natural Science Foundation of China (Grant No.62376267), the 2035 project of CASIA, the innoHK project, and the Strategic Priority Research Program of Chinese Academy of Sciences under Grant XDA0480200.

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

# A. Dataset Statistics

Table 8. Dataset Statistics for ICLR

| | | | | ICLR | | | | | Total |
|---|---|---|---|---|---|---|---|---|---|
| | 2017 | 2018 | 2019 | 2020 | 2021 | 2022 | 2023 | 2024 | |
| # papers | 487 | 907 | 1,550 | 2,210 | 2,590 | 2,659 | 3,821 | 5,703 | 19,927 |
| # tokens per paper | 7,666 | 8,762 | 9,553 | 10,197 | 11,354 | 12,877 | 12,974 | 21,830 | 12,206 |
| # reviews | 1,489 | 2,739 | 4,719 | 6,712 | 10,010 | 10,401 | 14,480 | 22,064 | 72,614 |
| # tokens per review | 411 | 514 | 557 | 541 | 639 | 812 | 793 | 686 | 635 |
| # weaknesses | 3,024 | 6,009 | 10,406 | 14,099 | 17,206 | 16,151 | 23,435 | 35,169 | 125,499 |
| # weaknesses per paper | 6.21 | 6.63 | 6.71 | 6.38 | 6.64 | 6.07 | 6.13 | 6.17 | 6.30 |
| # tokens per weakness | 20.42 | 21.09 | 21.02 | 21.41 | 21.51 | 21.04 | 21.01 | 21.82 | 21.02 |
| # strengths | 2,050 | 3,825 | 6,510 | 9,739 | 12,229 | 11,883 | 17,692 | 27,244 | 91,172 |
| # strengths per paper | 4.21 | 4.22 | 4.20 | 4.41 | 4.72 | 4.47 | 4.63 | 4.78 | 4.58 |
| # tokens per strength | 17.42 | 17.87 | 17.87 | 18.60 | 18.73 | 18.34 | 17.96 | 19.17 | 18.57 |
| % accepted | 41% | 37% | 32% | 31% | 33% | 41% | 41% | 40% | 37% |

Table 9. Dataset Statistics for NeurIPS

| | | | | | NeurIPS | | | | | Total |
|---|---|---|---|---|---|---|---|---|---|---|
| | 2016 | 2017 | 2018 | 2019 | 2020 | 2021 | 2022 | 2023 | 2024 | |
| # papers | 554 | 666 | 986 | 1,386 | 1,895 | 2,449 | 2,817 | 3,395 | 4,237 | 18,385 |
| # tokens per paper | 6,941 | 7,237 | 7,266 | 7,405 | 7,733 | 8,347 | 8,372 | 24,428 | 28,643 | 15,723 |
| # reviews | 3,159 | 1,938 | 3,008 | 4,243 | 7,261 | 9,589 | 10,393 | 15,175 | 16,647 | 71,413 |
| # tokens per review | 454 | 438 | 542 | 455 | 699 | 766 | 733 | 736 | 629 | 665 |
| # weaknesses | 3,367 | 4,002 | 6,137 | 8,758 | 11,027 | 14,125 | 16,982 | 19,239 | 26,079 | 109,716 |
| # weaknesses per paper | 6.08 | 6.01 | 6.22 | 6.32 | 5.82 | 5.77 | 6.03 | 5.67 | 6.16 | 5.97 |
| # tokens per weakness | 22.14 | 22.80 | 23.04 | 22.96 | 22.63 | 22.24 | 21.20 | 22.08 | 21.78 | 21.79 |
| # strengths | 2,871 | 3,367 | 5,314 | 7,429 | 10,123 | 12,148 | 13,975 | 17,093 | 22,199 | 94,519 |
| # strengths per paper | 5.18 | 5.06 | 5.39 | 5.36 | 5.34 | 4.96 | 4.96 | 5.03 | 5.24 | 5.14 |
| # tokens per strength | 19.17 | 19.61 | 19.88 | 18.99 | 18.88 | 19.24 | 18.40 | 18.99 | 19.35 | 18.90 |
| % accepted | 100% | 100% | 100% | 100% | 100% | 95% | 95% | 95% | 95% | 96% |

# B. Experiments Details

## B.1. Evaluation Metrics

### B.1.1. STRENGTHS-WEAKNESSES ANALYSIS METRICS

To evaluate the similarity between strengths and weaknesses generated by LLMs and those from human reviews, we compute pairwise similarity scores and use a similarity threshold $T$ to determine matches. Based on this, we define the following metrics(total refers to the combined strengths and weaknesses):

**Recall**: Measures the proportion of human review strengths or weaknesses that are captured by the LLM-generated reviews. A strength or weakness is considered "recalled" if at least one LLM-generated strength or weakness has a similarity score $\geq T$ with it.

$$\text{Recall}_{\text{Str.}} = \frac{\text{Recalled Human Strengths}}{\text{Total Human Strengths}}, \quad \text{Recall}_{\text{Wk.}} = \frac{\text{Recalled Human Weaknesses}}{\text{Total Human Weaknesses}}, \quad (1)$$

$$\text{Recall}_{\text{Total}} = \frac{\text{Total Recalled}}{\text{Total Human Strengths + Weaknesses}} \quad (2)$$

**Precision**: Measures the proportion of LLM-generated strengths or weaknesses that align with human-reviewed strengths or weaknesses. A generated strength or weakness is considered "precise" if at least one human-reviewed strength or weakness has a similarity score $\geq T$ with it.

$$\text{Precision}_{\text{Str.}} = \frac{\text{Precise LLM Strengths}}{\text{Total LLM Strengths}}, \quad \text{Precision}_{\text{Wk.}} = \frac{\text{Precise LLM Weaknesses}}{\text{Total LLM Weaknesses}}, \tag{3}$$

$$\text{Precision}_{\text{Total}} = \frac{\text{Total Precise}}{\text{Total LLM Strengths + Weaknesses}} \tag{4}$$

**F1-Score**: Balances Recall and Precision by computing their harmonic mean:

$$\text{F1-Score}_{\text{Str.}} = 2 \times \frac{\text{Precision}_{\text{Str.}} \times \text{Recall}_{\text{Str.}}}{\text{Precision}_{\text{Str.}} + \text{Recall}_{\text{Str.}}}, \quad \text{F1-Score}_{\text{Wk.}} = 2 \times \frac{\text{Precision}_{\text{Wk.}} \times \text{Recall}_{\text{Wk.}}}{\text{Precision}_{\text{Wk.}} + \text{Recall}_{\text{Wk.}}}, \tag{5}$$

$$\text{F1-Score}_{\text{Total}} = 2 \times \frac{\text{Precision}_{\text{Total}} \times \text{Recall}_{\text{Total}}}{\text{Precision}_{\text{Total}} + \text{Recall}_{\text{Total}}} \tag{6}$$

**MaxSim**: Measures the average maximum similarity between human-reviewed strengths or weaknesses and LLM-generated strengths or weaknesses. For each human strength or weakness, the highest similarity with any LLM-generated item is computed, and the average across all human-reviewed items is taken:

$$\text{MaxSim} = \frac{1}{N} \sum_{i=1}^{N} \max_j \text{Sim}(H_i, G_j) \tag{7}$$

where $N$ is the total number of strengths or weaknesses in the human review, $H_i$ represents the $i$-th human-reviewed strength or weakness, $G_j$ represents the $j$-th LLM-generated strength or weakness, and $\text{Sim}(H_i, G_j)$ is their similarity score.

**Jaccard Index**: Measures the overlap between human-reviewed and LLM-generated strengths or weaknesses(items can refer to strengths, weaknesses, or the total):

$$\text{Intersection} = \frac{\text{Recalled LLM Items + Precise Human Items}}{2} \tag{8}$$

$$\text{Jaccard} = \frac{\text{Intersection}}{\text{Total Human Items + Total LLM Items} - \text{Intersection}} \tag{9}$$

These metrics comprehensively evaluate the alignment between LLM-generated and human-reviewed strengths and weaknesses, capturing both coverage and accuracy, as well as their overall overlap.

### B.1.2. DECISIONS ANALYSIS METRICS

To address potential bias caused by class imbalance, we supplemented Accuracy with the F1-score, Matthews Correlation Coefficient (MCC), Balanced Accuracy (Bal. Acc), and G-mean. These metrics provide greater robustness in imbalanced scenarios by accounting for both positive and negative class distributions.

**Definitions:**

- **TP (True Positives)**: Cases where the LLM correctly predicts acceptance, matching the human reviewer's decision.
- **TN (True Negatives)**: Cases where the LLM correctly predicts rejection, matching the human reviewer's decision.
- **FP (False Positives)**: Cases where the LLM predicts acceptance, but the human reviewer rejects.
- **FN (False Negatives)**: Cases where the LLM predicts rejection, but the human reviewer accepts.

Based on these definitions, the metrics are calculated as follows:

$$\text{Accuracy} = \frac{TP + TN}{TP + TN + FP + FN} \tag{10}$$

$$\text{F1-Score} = 2 \times \frac{\text{Precision} \times \text{Recall}}{\text{Precision} + \text{Recall}}, \quad \text{where} \quad \text{Precision} = \frac{TP}{TP + FP}, \quad \text{Recall} = \frac{TP}{TP + FN} \tag{11}$$

$$\text{MCC} = \frac{TP \times TN - FP \times FN}{\sqrt{(TP + FP)(TP + FN)(TN + FP)(TN + FN)}} \tag{12}$$

$$\text{Bal. Acc} = \frac{1}{2} \left( \frac{TP}{TP + FN} + \frac{TN}{TN + FP} \right) \tag{13}$$

$$\text{G-mean} = \sqrt{\frac{TP}{TP + FN} \times \frac{TN}{TN + FP}} \tag{14}$$

These metrics collectively evaluate the model's performance under imbalanced conditions by considering both the correctness of predictions and the balance between positive and negative classes.

### B.2. Compared Methods Configurations

We followed the default settings for each method and extracted strengths and weaknesses from their review comments for evaluation. For AgentReview, we adopted the baseline settings of the method. To enable strengths-weaknesses evaluation, we used GPT-4o-mini to extract strengths and weaknesses from the AC's final review. For AI-Scientist, we followed its default settings, including `num_reflections=5`, `num_fs_examples=1`, `num_reviews_ensemble=5`, and `temperature=0.1`. Since its reviews contain formatted strengths and weaknesses, we directly used them for evaluation. For LLM Review, we parsed the paper content, including the title, abstract, and captions of tables and figures, to construct the standardized input as specified by the method. We then used GPT-4o-mini to extract strengths and weaknesses from its complete review for evaluation.

### B.3. Detailed Main Results

Tables 10 and 11 present detailed strengths-weaknesses and decision analysis results. We compare Agent Reviewers with existing methods and a *Single Agent* method, which uses a single text-only agent to directly generate the final comments and decision. Table 10 shows that Agent Reviewers perform well across all metrics, particularly balancing coverage and accuracy (see F1-score and Jaccard). Table 11 shows that, compared to other methods, Agent Reviewers achieve the highest alignment with human reviewers in decision-making.

*Table 10.* Detailed strengths-weaknesses analysis main results. Str. denotes strengths, and Wk. denotes weaknesses. Recall and F1-score are calculated using a similarity threshold of 0.5, chosen for its alignment with human perception of similarity.

| Method | LLM | Recall↑ | | | F1-score↑ | | | MaxSim↑ | | | Jaccard↑ | | |
|---|---|---|---|---|---|---|---|---|---|---|---|---|---|
| | | Total | Str. | Wk. | Total | Str. | Wk. | Total | Str. | Wk. | Total | Str. | Wk. |
| AgentReview (Jin et al., 2024b) | GPT-4o-mini | 0.314 | 0.445 | 0.209 | 0.340 | 0.442 | 0.245 | 0.438 | 0.495 | 0.391 | 0.215 | 0.252 | 0.194 |
| AI_Scientist (Lu et al., 2024) | GPT-4o-mini | 0.361 | 0.453 | 0.287 | 0.426 | 0.523 | 0.345 | 0.444 | 0.477 | 0.418 | 0.285 | 0.313 | 0.272 |
| LLM Review (Liang et al., 2024) | GPT-4o-mini | **0.439** | 0.520 | **0.373** | 0.420 | 0.560 | 0.328 | **0.487** | 0.515 | **0.466** | 0.275 | 0.243 | **0.336** |
| Single Agent | GPT-4o-mini | 0.357 | 0.496 | 0.245 | 0.430 | 0.575 | 0.302 | 0.444 | 0.488 | 0.410 | 0.284 | 0.317 | 0.268 |
| Agent Reviewers | GPT-4o-mini | 0.418 | **0.541** | 0.319 | **0.462** | **0.577** | **0.357** | 0.477 | **0.526** | 0.438 | **0.315** | **0.333** | 0.310 |

*Table 11.* Detailed decisions analysis main results. AgentReview(Top-$k$) accepts the top-$k$ ranked papers per batch of 10 papers.

| Method | LLM | Decision Offered? | Acc↑ | F1-score↑ | MCC↑ | Bal. Acc↑ | G-mean↑ |
|---|---|---|---|---|---|---|---|
| AgentReview(Top-3)(Jin et al., 2024b) | GPT-4o-mini | ✓ | 0.587 | 0.410 | 0.104 | 0.549 | 0.515 |
| AgentReview(Top-4)(Jin et al., 2024b) | GPT-4o-mini | ✓ | 0.533 | 0.417 | 0.028 | 0.514 | 0.505 |
| AI_Scientist(Lu et al., 2024) | GPT-4o-mini | ✓ | **0.610** | 0.049 | 0.123 | 0.513 | 0.158 |
| LLM Review(Liang et al., 2024) | GPT-4o-mini | ✗ | - | - | - | - | - |
| Single Agent | GPT-4o-mini | ✓ | **0.610** | 0.489 | 0.175 | 0.586 | 0.574 |
| Agent Reviewers | GPT-4o-mini | ✓ | 0.607 | **0.566** | **0.220** | **0.613** | **0.612** |

## B.4. Detailed Ablation Results

Table 12 presents the performance of Agent Reviewers across different LLMs, including GPT-4o-mini, Gemini-exp-1206, and Deepseek-V3-1226. The experiments were conducted on the same set of 300 ICLR 2024 papers used in the main study. We compare it with a single agent method, which uses a single text-based agent to directly generate the final review. Notably, the knowledge cutoff dates for GPT-4o-mini and Gemini-exp-1206 precede the public release of ICLR 2024, ensuring no data leakage. However, DeepSeek-V3 has a later knowledge cutoff, posing a potential risk of knowledge leakage. Nevertheless, our experiments did not reveal any anomalously high performance suggestive of data leakage, nor any clear signs of memorization of paper content or reviews. Therefore, we included it in our evaluation. The results indicate that Agent Reviewers achieve consistently strong performance across different LLMs, showing improvements over the single-agent method.

*Table 12.* Detailed experiments on different LLMs. *Single Agent* means using a single text-only agent to generate review comments directly.[†]Using GPT-4o-mini for multimodal ability, as DeepSeek-V3 is text-only at the time of writing.

| LLM | Method | Fine-grained Review Analysis | | | | | | Acceptance Decisions Analysis | | | |
|---|---|---|---|---|---|---|---|---|---|---|---|
| | | F1-score↑ | | | Jaccard↑ | | | F1-score↑ | MCC↑ | Bal. Acc↑ | G-mean↑ |
| | | Total | Str. | Wk. | Total | Str. | Wk. | | | | |
| GPT-4o-mini | Single Agent | 0.430 | 0.575 | 0.302 | 0.284 | 0.317 | 0.268 | 0.489 | 0.175 | 0.586 | 0.574 |
| | Agent Reviewers | 0.462 | 0.577 | 0.357 | 0.315 | 0.333 | 0.310 | 0.566 | 0.220 | 0.613 | 0.612 |
| | *Improvement* | +0.032 | +0.002 | +0.055 | +0.031 | +0.016 | +0.042 | +0.077 | +0.045 | +0.027 | +0.038 |
| Gemini-exp-1206 | Single Agent | 0.467 | 0.606 | 0.345 | 0.325 | 0.321 | 0.340 | 0.308 | 0.092 | 0.542 | 0.493 |
| | Agent Reviewers | 0.465 | 0.603 | 0.355 | 0.316 | 0.284 | 0.364 | 0.452 | 0.113 | 0.556 | 0.542 |
| | *Improvement* | -0.002 | -0.003 | +0.010 | -0.007 | -0.037 | +0.024 | +0.144 | +0.021 | +0.014 | +0.049 |
| Deepseek-V3-1226[†] | Single Agent | 0.427 | 0.583 | 0.289 | 0.288 | 0.300 | 0.288 | 0.578 | 0.097 | 0.519 | 0.234 |
| | Agent Reviewers | 0.485 | 0.602 | 0.383 | 0.329 | 0.293 | 0.379 | 0.589 | 0.155 | 0.551 | 0.390 |
| | *Improvement* | +0.057 | +0.019 | +0.094 | +0.041 | -0.007 | +0.091 | +0.010 | +0.058 | +0.032 | +0.156 |

## B.5. Experiments on Using Different LLMs for Domain-Specific Reviewers

We consider the diversity of domain-specific reviewers to be an important factor in the review process. To further explore this, we experimented with assigning different large language models (LLMs) to different domain-specific reviewers, as shown in Table 13. Specifically, we used GPT-4o-mini, Gemini-exp-1206, and Deepseek-V3-0324 as the LLMs for three domain-specific reviewers, and evaluated them on the same set of 100 randomly sampled papers. The results show that the multi-LLM setup did not yield performance improvements. We attribute this to the fact that our system already encourages diverse perspectives by equipping each reviewer with distinct domain knowledge, leaving limited room for further gains through LLM heterogeneity.

## B.6. Evaluation across Conferences and Years

Table 14 explores the review performance across conferences(ICLR 2024 and NeurIPS 2024). We evaluate the proposed Agent Reviewers against a Single Agent baseline, where a single text-only agent directly generates the final review. For each conference, 100 papers were randomly sampled for evaluation using the default language model, GPT-4o-mini. The results across both conferences are consistent: Agent Reviewers consistently outperform the Single Agent baseline, with an average relative improvement of 23.5% on ICLR and 44.7% on NeurIPS.

*Table 13.* Comparison between using a single LLM and multiple LLMs. [†]Using GPT-4o-mini for multimodal ability, as DeepSeek-V3 is text-only at the time of writing.

| LLM | Strengths-Weaknesses Analysis | | | | | | Decisions Analysis | | | |
|---|---|---|---|---|---|---|---|---|---|---|
| | F1-score↑ | | | Jaccard↑ | | | F1-score↑ | MCC↑ | Bal. Acc↑ | G-mean↑ |
| | Total | Str. | Wk. | Total | Str. | Wk. | | | | |
| GPT-4o-mini | 0.453 | 0.558 | **0.361** | **0.307** | **0.333** | 0.296 | **0.605** | **0.385** | **0.705** | **0.705** |
| Gemini-exp-1206 | 0.430 | 0.561 | 0.327 | 0.289 | 0.283 | 0.304 | 0.536 | 0.367 | 0.668 | 0.638 |
| Deepseek-V3-0324[†] | **0.458** | 0.564 | **0.370** | 0.306 | 0.288 | **0.332** | 0.552 | 0.299 | 0.618 | 0.485 |
| Multi-LLMs | 0.454 | **0.568** | 0.356 | 0.304 | 0.303 | 0.313 | 0.514 | 0.241 | 0.628 | 0.627 |

*Table 14.* Comparison of review performance across conferences (ICLR 2024 and NeurIPS 2024). *Single Agent* means using a single text-only agent to generate review comments directly.

| Conference | Method | Strengths-Weaknesses Analysis | | | | Decisions Analysis | | | |
|---|---|---|---|---|---|---|---|---|---|
| | | Recall | F1 | MaxSim | Jaccard | F1 | MCC | Bal.A | G-mean |
| ICLR 2024 | Single Agent | 0.347 | 0.416 | 0.442 | 0.270 | 0.469 | 0.219 | 0.609 | 0.593 |
| | Agent Reviewers | 0.409 | 0.453 | 0.476 | 0.307 | 0.605 | 0.385 | 0.705 | 0.705 |
| | *Improvement* | 0.062 | 0.037 | 0.034 | 0.037 | 0.136 | 0.166 | 0.096 | 0.112 |
| NeurIPS 2024 | Single Agent | 0.357 | 0.440 | 0.444 | 0.299 | 0.349 | 0.039 | 0.525 | 0.424 |
| | Agent Reviewers | 0.418 | 0.461 | 0.476 | 0.319 | 0.589 | 0.120 | 0.591 | 0.569 |
| | *Improvement* | 0.061 | 0.021 | 0.032 | 0.020 | 0.240 | 0.081 | 0.066 | 0.145 |

Table 15 explores the review performance across years(ICLR 2024 and ICLR 2023). We evaluate the proposed Agent Reviewers against a Single Agent baseline, where a single text-only agent directly generates the final review. For each conference, 100 papers were randomly sampled for evaluation using the default language model, GPT-4o-mini. The cutoff year for the shared memory pool (SMP) was set to one year prior to the test data year, to avoid data leakage from SMP. It is worth noting that, in the ICLR 2023 evaluation, GPT-4o-mini's knowledge cutoff postdates the paper release, posing a potential risk of data leakage—though no clear signs of memorization were observed. Whether in 2023 or 2024, Agent Reviewers have shown better results than the single-agent baseline, demonstrating the consistency of the gains of our method. For ICLR 2023, the size of the shared memory pool (SMP)—an essential component of our method—was reduced from 28,372 to 21,156 papers (a 25% decrease) due to the earlier cutoff year. This reduction may have limited the effectiveness of decision analysis compared to ICLR 2024.

*Table 15.* Comparison of Review Performance across Years(ICLR 2024 and ICLR 2023). *Single Agent* means using a single text-only agent to generate review comments directly.

| Conference | Method | Strengths-Weaknesses Analysis | | | | Decisions Analysis | | | |
|---|---|---|---|---|---|---|---|---|---|
| | | Recall | F1 | MaxSim | Jaccard | F1 | MCC | Bal.A | G-mean |
| ICLR 2024 | Single Agent | 0.347 | 0.416 | 0.442 | 0.270 | 0.469 | 0.219 | 0.609 | 0.593 |
| | Agent Reviewers | 0.409 | 0.453 | 0.476 | 0.307 | 0.605 | 0.385 | 0.705 | 0.705 |
| | *Improvement* | 0.062 | 0.037 | 0.034 | 0.037 | 0.136 | 0.166 | 0.096 | 0.112 |
| ICLR 2023 | Single Agent | 0.334 | 0.408 | 0.432 | 0.271 | 0.474 | 0.169 | 0.580 | 0.554 |
| | Agent Reviewers | 0.390 | 0.439 | 0.467 | 0.298 | 0.538 | 0.139 | 0.570 | 0.570 |
| | *Improvement* | 0.056 | 0.031 | 0.035 | 0.027 | 0.064 | -0.030 | -0.010 | 0.016 |

## B.7. Detailed Strengths-Weaknesses Analysis

We observe in Table 2 that nearly all methods are less effective at identifying paper weaknesses compared to strengths. We hypothesize that this discrepancy arises because the strengths proposed by reviewers tend to be relatively consistent, e.g., similar to claimed contributions, whereas weaknesses are more diverse and often depend on the reviewers' individual background and subjective judgment. This diversity makes it more difficult for LLM-generated weaknesses to match humans'. To further investigate this, we analyzed the semantic similarity of strengths and weaknesses generated by Agent Reviewers using GPT-4o-mini, Gemini, and Deepseek-V3 for the same set of papers. On average, the similarity scores between GPT-4o-mini and Gemini were 0.664 for strengths and 0.484 for weaknesses, while the similarity between GPT-4o-mini and Deepseek-V3 was 0.738 for strengths and 0.663 for weaknesses.

We present a classification of the strengths and weaknesses identified by Agent Reviewers (using GPT-4o-mini) and human reviewers on the subset of 300 papers from ICLR 2024 in Figure 6. We used GPT-4o-mini to classify the strengths and

weaknesses, with each strength or weakness being classified into up to two categories. In terms of strengths, both Agent Reviewers and human reviewers placed significant emphasis on "Implications of the research" and "Novelty," with a high degree of alignment between the two. However, Agent Reviewers mentioned "Reproducibility" far less frequently compared to human reviewers. Regarding weaknesses, Agent Reviewers mentioned "Clarity and presentation" more frequently than human reviewers, but they fell short in addressing aspects such as "Comparison with related work," "Theoretical soundness," and "Novelty." We believe that this detailed analysis can help us understand in which areas the evaluations of Agent Reviewers may align more closely with those of human reviewers, thereby indicating areas where their assessments may hold greater value.

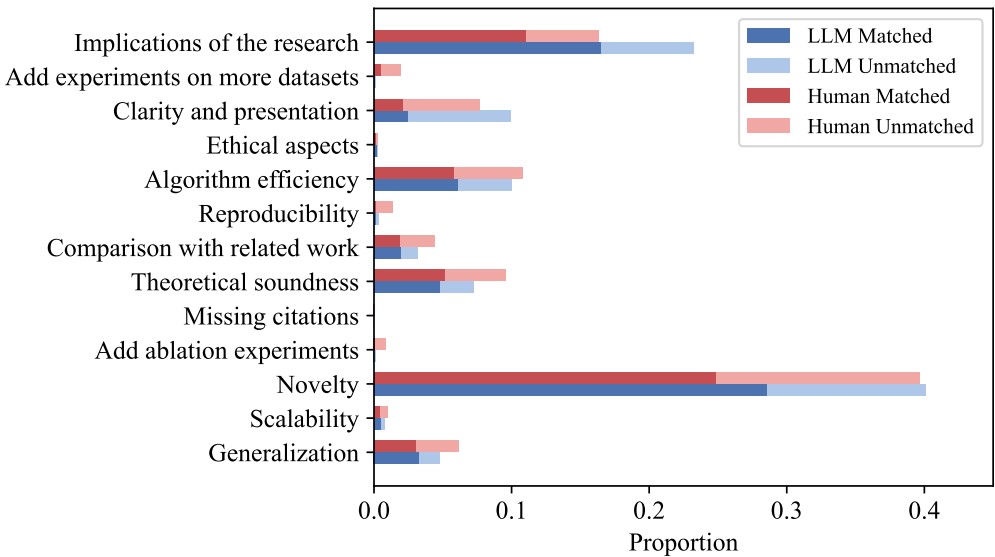

(a) Proportion of different opinion categories in LLM/Human strengths(stacked matched/unmatched)

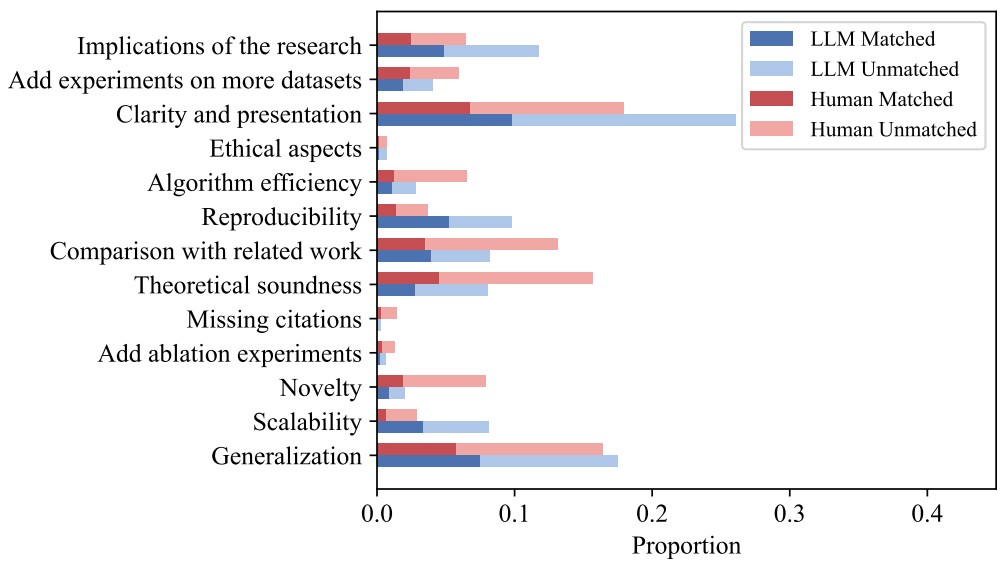

(b) Proportion of different opinion categories in LLM/Human weaknesses(stacked matched/unmatched)

*Figure 6.* Proportion of different opinion categories in LLM/Human strengths and weaknesses (stacked matched/unmatched). Proportion refers to the ratio of the number of strengths/weaknesses to the total number of strengths/weaknesses.

## C. Case Study

Tables 16 and 17 present the strengths and weaknesses generated by Agent Reviewers for *KAN: Kolmogorov–Arnold Networks* and *LoRA: Low-Rank Adaptation of Large Language Models*, using Gemini-exp-1206. Agent Reviewers produce insightful, detailed, and comprehensive critiques, incorporating discussions on the specific previous works and analyses of figures. Tables 18 present a case study on our own paper, where Agent Reviewers provide constructive and insightful feedback.

*Table 16.* A case study on *KAN: Kolmogorov–Arnold Networks*, demonstrating weaknesses and strengths generated by Agent Reviewers.

| KAN: Kolmogorov–Arnold Networks | |
|---|---|
| **Weaknesses** | 🙁 The paper does not sufficiently leverage or discuss existing research on learnable activation functions (e.g., ICLR 2017, ICLR 2021, ICLR 2022) or related concepts, limiting the context and perceived novelty of the proposed approach. |
| | 🙁 The paper does not leverage the findings from previous research papers about 'Kolmogorov-Arnold Networks (KANs)', such as 'ICLR_2023_3364' which proposed a novel theoretical architecture, or 'ICLR_2022_1438' which applied Koopman operator theory to neural sequential models. |
| | 🙁 While KANs are shown to outperform MLPs in specific tasks, the paper does not adequately address potential limitations in scalability or handling complex, high-dimensional data, despite acknowledging that KANs typically require smaller computation graphs than MLPs. |
| | 🙁 The empirical validation is primarily limited to small-scale AI + Science tasks, and the paper lacks extensive experimental results on a wider range of datasets to demonstrate the generalizability of KANs. |
| | 🙁 The paper does not provide a comprehensive comparison with other state-of-the-art methods beyond MLPs, making it difficult to assess the relative performance of KANs in a broader context, especially regarding computational cost and efficiency. |
| | 🙁 The practical implementation details, potential challenges, and computational overhead of using KANs in real-world applications are not extensively discussed. |
| | ... |
| **Strengths** | 🙂 The paper introduces Kolmogorov-Arnold Networks (KANs) as a novel alternative to Multi-Layer Perceptrons (MLPs), based on the Kolmogorov-Arnold representation theorem, representing a significant shift in neural network architecture. |
| | 🙂 KANs utilize learnable activation functions on edges, parameterized as splines, instead of fixed activation functions on nodes, offering enhanced flexibility and expressiveness in function approximation. |
| | 🙂 KANs are presented as more interpretable than MLPs, with visualizations (Figures 0.1, 2.1, 2.2) illustrating their structure and potential for use in scientific discovery, aiding in the (re)discovery of mathematical and physical laws. |
| | 🙂 The paper demonstrates that smaller KANs can achieve comparable or better accuracy than larger MLPs in function fitting tasks, as supported by results in mathematics and physics examples. |
| | ... |

*Table 17.* A case study on *LoRA: Low-Rank Adaptation of Large Language Models*, demonstrating weaknesses and strengths generated by Agent Reviewers.

| LoRA: Low-Rank Adaptation of Large Language Models |
|---|

**Weaknesses** 🙁 The paper's exploration of the relationship between rank and performance is limited, particularly in how the optimal rank is determined for different tasks and models. A more systematic study on rank selection criteria would be beneficial.

🙁 The paper notes that the performance of adaptation methods does not always monotonically improve with more trainable parameters, as shown in Figure 2, which could be seen as a limitation or area for further investigation. A deeper exploration of this phenomenon would strengthen the paper.

🙁 The paper does not explore the potential synergies or trade-offs with methods like those proposed in other papers, which focuses on memory footprint reduction. Such an analysis would provide a more comprehensive understanding of LoRA's capabilities.

🙁 The paper does not provide a detailed comparison with other parameter-efficient methods like Compacter, which could highlight the relative advantages and disadvantages of LoRA, especially in terms of performance and computational efficiency.

🙁 The paper acknowledges that it's not straightforward to batch inputs to different tasks when using LoRA, which could limit its applicability in multi-task scenarios. A more thorough discussion and potential solutions to this limitation are needed.

...

**Strengths** 🙂 Introduces LoRA, a novel parameter-efficient adaptation method that significantly reduces the number of trainable parameters by freezing pre-trained weights and injecting trainable rank decomposition matrices.

🙂 LoRA is orthogonal to many prior methods and can be combined with them, enhancing its flexibility and potential for further improvements, as demonstrated in Appendix E.

🙂 The linear design of LoRA allows merging the trainable matrices with the frozen weights when deployed, introducing no inference latency compared to a fully fine-tuned model.

🙂 Provides empirical evidence that LoRA can reduce the number of trainable parameters by up to 10,000 times and GPU memory requirements by 3 times compared to fine-tuning GPT-3 175B with Adam.

🙂 Demonstrates that LoRA performs on par with or better than full fine-tuning on various models (RoBERTa, DeBERTa, GPT-2, GPT-3) while having fewer trainable parameters and higher training throughput.

...

*Table 18.* A case study on this paper, using Deepseek-V3.

| Agent Reviewers: Domain-specific Multimodal Agents with Shared Memory for Paper Review |
|---|

**Keywords and memory**

**Agent Reviewers**
- You Are the Best Reviewer of Your Own Papers: An Owner-Assisted Scoring Mechanism
- Group Fairness in Peer Review
- A One-Size-Fits-All Approach to Improving
- Randomness in Paper Assignment
- Counterfactual Evaluation of Peer-Review Assignment Policies

**Shared Memory Pool**
- Mention Memory: incorporating textual knowledge into Transformers through entity mention attention
- An Investigation into the Role of Author Demographics in ICLR Participation and Review
- What's new? Summarizing Contributions in Scientific Literature
- SciRepEval: A Multi-Format Benchmark for Scientific Document Representations
- Sparse is Enough in Scaling Transformers

**Multimodal Agent**
- You Are the Best Reviewer of Your Own Papers: An Owner-Assisted Scoring Mechanism
- Perceptual Score: What Data Modalities Does Your Model Perceive?
- Humanly Certifying Superhuman Classifiers
- Teaching Machines to Describe Images with Natural Language Feedback
- Paraphrasing Is All You Need for Novel Object Captioning

**Comments**

**Weaknesses**

☹ The reliance on pre-trained LLMs and multimodal models may limit the generalizability of the system, especially in domains with limited or no pre-existing data.

☹ There may be potential gaps in the experimental design, particularly in how the performance of the proposed system is benchmarked against existing methods. The paper lacks a detailed discussion on the evaluation metrics and datasets used.

☹ The paper does not fully address potential scalability issues, especially given the large volume of submissions in conferences like ICLR. It is unclear how the system would perform under high-load conditions.

☹ The generalizability of the proposed system to other domains or conferences has not been thoroughly explored, raising concerns about its applicability beyond the tested scenarios.

...

**Strengths**

☺ The system leverages both text and visual elements (e.g., figures, layout) to provide more comprehensive feedback, addressing a significant limitation of existing AI-based review systems.

☺ The shared memory pool (SMP) is an innovative feature that mimics real reviewers' background knowledge, enhancing the system's ability to generate domain-specific and insightful feedback.

☺ The results indicate significant performance improvements in paper review tasks, as shown in the comparative analysis with existing systems, demonstrating the practical applicability of the proposed approach.

☺ The system architecture, including Multi-agent Interaction (MI), Shared Memory Pool (SMP), and Multimodal Agent (MA), is well-designed and aligns with real-world peer review practices.

☺ The paper presents a well-structured layout with clear sections, effective use of headings, and high-quality figures that enhance understanding of the content.

...

**Decision**

**score**      6

**recommendation**      Accept

# D. Prompts

## D.1. Prompts for dataset standardization

---

Prompt for Review Summarization

```
You are an expert in computer science and artificial intelligence. I will provide
you with multiple reviews of a machine learning conference paper. Each review begins
 with "<|review_start|>" and ends with "<|review_end|>":
% REVIEWS:
{reviews}

Your task is to generate a final summary in JSON format based on these reviews. The
JSON object should contain two keys: 'strengths' and 'weaknesses', each mapping to a
 list. The 'strengths' list should capture the positive aspects of the paper as
potential reasons for acceptance, while the 'weaknesses' list should capture the
negative aspects, questions, or suggestions as potential reasons for rejection.

Please ensure that:
- Each item in the lists is specific, detailed, and avoids generalization.
- The items are not duplicated.
- All key points from the original reviews are covered.
- The summary does not introduce any content that is not present in the original
reviews.

Only output the JSON content, and do not include any additional text or commentary,
as it will not be parsed correctly. Your output should be formatted as a markdown
code snippet according to the following pattern:

```json
{
  "strengths": [String],
  "weaknesses": [String]
}
```

---

## D.2. Prompts for Agent Reviewers

---

Prompt for Mulitmodal reviewer agent

---

```
As a multimodal agent, your task is to act as an AI expert, and provide valuable
information and evaluations to assist in reviewing an AI conference paper, based
solely on a **thumbnail** of the paper. Please adhere to the following requirements:
- **Avoid Repeating the Abstract**: Do not include information that overlaps with
the paper's abstract. Focus on content not covered in the abstract.
- **Base on Visible Information**: All content must strictly derive from the
information visible in the thumbnail. Avoid any form of speculation or assumption.
- **Carefully Avoid Hallucinations**: Only describe what you can directly observe in
 the thumbnail. Do not add uncertain details.
- **Use Structured Output**: Organize your response in a clear and orderly manner to
 facilitate understanding and reference by other agent.
- **Clear and Concise Language**: Use precise and straightforward language to ensure
 accurate information delivery.
Please provide your output in the following JSON format, starting with '{' and
ending with '}' followed by "<|END_OF_JSON|>":
{
    "strengths": {
        "innovation": "Describe the innovative aspects in research methods,
        theoretical frameworks, or experimental results.",
        "visual_presentation": "Evaluate the paper's formatting, quality of figures
        and tables, and overall visual appeal."
    },
    "weaknesses": {
        "methodological_issues": "Point out any potential shortcomings in experimental
         design, data analysis, or theoretical reasoning.",
        "writing_quality": "Assess the language expression, logical structure, and
        readability of the paper."
    },
    "additional_key_findings": {
        "results_interpretation": "Provide interpretations of key results based on the
         charts and data in the thumbnail, avoiding repetition of the abstract.",
        "theoretical_or_practical_significance": "Discuss the impact or application
        value of these results in the relevant field."
    },
    "structure_and_organization": {
        "section_layout": "Describe the paper's section arrangement and content
        distribution to help the B-Group understand the overall framework.",
        "figures_and_appendices": "Mention important figures, tables, and appendices,
        and their roles in the paper."
    },
    "overall_evaluation_and_suggestions": {
        "comprehensive_evaluation": "Provide an objective assessment of the paper's
        overall quality based on the above information.",
        "publication_suggestions": "Suggest whether the paper is suitable for
        publication or what modifications are needed."
    }
}

**Important Notes**:

- **Strictly Based on the Thumbnail**: All information must originate from the
content visible in the thumbnail. Avoid subjective guesses.
- **Avoid Abstract Repetition**: Focus on providing details and insights not
mentioned in the abstract.
- **Maintain Objectivity**: Provide impartial evaluations without personal bias.
- **Clear and Specific Information**: Ensure each piece of information is clear and
verifiable.

Please follow the above guidelines to complete your task and assist the other agents
 in conducting a high-quality review without access to the paper's visual content.
```

Prompt for Meta-Reviewer to generate keywords

```
You are an expert in the field of machine learning. Based on the provided paper
content and the following thumbnail information from a vision agent, extract {
expected_count} key topics or keywords that best represent the core ideas of the
paper. Ensure that these keywords are distinct, relevant, and represent the most
critical aspects of the work.

Thumbnail Information: {json.dumps(thumbnail_info)}

If a keyword is synonymous with an existing keyword listed below, use the existing
one instead of creating a new one:
Existing Keywords: {existing_keywords_str}

Please provide the keywords in JSON format as a list of strings, without any
additional text or code block indicators.
Example: ["keyword1", "keyword2", "keyword3"]

Here is the paper content(extracted a part, may be incomplete):
<PAPER_BEGIN>
{paper_content}
<PAPER_END>

Output the keywords below:
```

---

Prompt for domain-specific agent(initial review)

---

You are an AI researcher tasked with reviewing a paper submitted to a prestigious
machine learning conference. Your objective is to critically evaluate the paper
according to the provided guidelines. Be thorough and cautious in your review. Pay
special attention to aspects related to "{keyword}". Based on the provided paper
content and the thumbnail information from a vision agent. Additionally, here is
some information from previous research papers related to the keyword: {
memory_insertion}

Here is the paper you are asked to review (extracted a part, may be incomplete):
<PAPER_BEGIN>
{paper_content}
<PAPER_END>

Here is the thumbnail information from a vision agent: {json.dumps(thumbnail_info)}

Provide only the JSON object with no extra formatting (e.g., no "```", "```json", or
 similar), starting with `{{` and ending with `}}` followed by "<|END_OF_JSON|>":

{{
"strengths": [
   "Clearly list and describe the key contributions or innovations made by the paper,
    with each strength on a separate line.",
   "Provide concrete examples or evidence that highlight the effectiveness of the
   approach."
],
"weaknesses": [
   "Identify and list any significant limitations or flaws, ensuring that each
   weakness is articulated on a separate line.",
   "Mention any concerns regarding scalability, robustness, or generalizability, and
    suggest areas for improvement."
],
"overall_evaluation": {{
   "justification": "Summarize your overall assessment, focusing on the balance
   between the strengths and weaknesses.",
   "score": "0-10",
   "recommendation": "Accept/Reject",
}}
}}

**Important:** Your recommendation must be either "Accept" or "Reject." Options like
 "Accept with Revisions" are not allowed! Ensure your review reflects a balanced
evaluation of strengths and weaknesses.

**Note:** The conference acceptance rate is 40%, please review carefully.

Begin your JSON response here and end with "<|END_OF_JSON|>".

---

Prompt for domain-specific agent(discussion period)

---

```
As an AI reviewer, your task is to revise your review focused on the keyword '{
keyword}'. You will be provided with your initial review, feedback from other
reviewers, relevant memory information about '{keyword}', the paper content, and
thumbnail information from a vision agent. Use these resources to refine and improve
 your review.

Below is your initial review:

{json.dumps(reviewer_feedback, indent=2)}

Consider the following review from other reviewers on different aspects of the paper:

{other_feedback}

{memory_insertion}

Here is the paper text (extracted part, may be incomplete):
<PAPER_BEGIN>
{paper_content}
<PAPER_END>

Here is the thumbnail information from a vision agent: {json.dumps(thumbnail_info)}

Please revise your review by taking into account the feedback from other reviewers,
the information about '{keyword}', and the original paper. Maintain your independent
 perspective while carefully evaluating and improving your review, addressing any
significant issues.

Provide only the JSON object with no extra formatting (e.g., no "```", "```json", or
 similar), starting with '{{' and ending with '}}' followed by "<|END_OF_JSON|>":

{{
   "strengths": [
      "Clearly list and describe the key contributions or innovations made by the
      paper, with each strength on a separate line.",
      "Provide concrete examples or evidence that highlight the effectiveness of the
       approach."
   ],
   "weaknesses": [
      "Identify and list any significant limitations or flaws, ensuring that each
      weakness is articulated on a separate line.",
      "Mention any concerns regarding scalability, robustness, or generalizability,
      and suggest areas for improvement."
   ],
   "overall_evaluation": {{
      "justification": "Summarize your overall assessment, focusing on the balance
      between the strengths and weaknesses.",
      "score": "0-10",
      "recommendation": "Accept or Reject"
   }}
}}
**Important:** Your recommendation must be either "Accept" or "Reject." Options like
 "Accept with Revisions" are not allowed! Ensure your review reflects a balanced
evaluation of strengths and weaknesses.

**Note:** The conference acceptance rate is 40%, please review carefully.

Begin your JSON response here and end with "<|END_OF_JSON|>".
```

---

Prompt for AC(final review)

---

You are an Area Chair (AC) at a prestigious AI conference. Your responsibility is to provide a meta-review for a paper that has already been reviewed by multiple AI reviewers with expertise in different fields. Each reviewer has provided their feedback in the form of a structured JSON object. You will receive the paper content and thumbnail information from a vision agent. Your task is to **consider and integrate the insights from these reviews to generate a comprehensive final review**. Ensure that **all reasonable strengths and weaknesses from each reviewer are covered**!

Here is the paper text (extracted a part, may be incomplete):
<PAPER_BEGIN>
{paper_content}
<PAPER_END>

Here is the thumbnail information from a vision agent: {json.dumps(thumbnail_info)}

Ensure that you consider each reviewer's strengths, weaknesses, and overall evaluation, and make a well-justified final recommendation. Your final review should be structured in the same JSON format provided below.

Here are the individual reviews:
{review_prompts}

Provide only the JSON object with no extra formatting (e.g., no "```", "```json", or similar), starting with `{{` and ending with `}}` followed by "<|END_OF_JSON|>":

```
{{
    "strengths": [
        "Clearly list and describe the key contributions or innovations made by the
        paper, with each strength on a separate line.",
        "Provide concrete examples or evidence that highlight the effectiveness of the
         approach."
        "**Ensure that all reasonable points from each reviewer are covered!**"
    ],
    "weaknesses": [
        "Identify and list any significant limitations or flaws, ensuring that each
        weakness is articulated on a separate line.",
        "Mention any concerns regarding scalability, robustness, or generalizability,
        and suggest areas for improvement."
        "**Ensure that all reasonable points from each reviewer are covered!**"
    ],
    "overall_evaluation": {{
        "justification": "Summarize your overall assessment, focusing on the balance
        between the strengths and weaknesses.",
        "score": "0-10",
        "recommendation": "Accept/Reject" # choose one from them
    }}
}}
```

**Important:** Please make sure:
1. Ensure that strengths and weaknesses should cover all reasonable points from each reviewer! Coverage is very important!
2. Again: Ensure that strengths and weaknesses should cover all reasonable points from each reviewer! Coverage is very important!
3. Your recommendation must be either "Accept" or "Reject." Options like "Accept with Revisions" are not allowed!
4. The conference acceptance rate is 40%, please review carefully.

Begin your JSON response here and end with "<|END_OF_JSON|>".".

## D.3. Prompts for SMP initialization

---

Prompt for keywords extraction

---

```
You are an expert in artificial intelligence. Based on the provided paper content,
extract {keywords_num} specific and distinct keywords that best represent the core
ideas of the paper-avoiding broad terms like "Artificial Intelligence", "Neural
Networks", or "Deep Learning"-and provide a concise summary of the paper.

Please format the output in JSON with the following structure(end with "<|
END_OF_JSON|>"):
<|START_OF_JSON|>
{{
    "keywords": [
        {{
            "keyword": "keyword_1",
            "description": "keyword_1 is xxxxxx."
        }},
        {{
            "keyword": "keyword_2",
            "description": "keyword_2 is xxxxxx."
        }},
        ...
    ],
    "summary": "A concise summary of the paper's content."
}}
<|END_OF_JSON|>

**Instructions:**
- Do not include any additional text outside the JSON structure.
- Ensure that the JSON is properly formatted without any syntax errors.
- The descriptions should follow the format "keyword is xxx." and provide an overall
 introduction to each keyword from the perspective of the entire field, not limited
to the paper.
- The summary should be a brief overview capturing the main points of the paper.

Below are the article's title, abstract, and introduction:
<PAPER_BEGIN>
{paper_content}
<PAPER_END>

Output the keywords and summary below(end with "<|END_OF_JSON|>"):
```

---

---

Prompt for keywords aggregation

---

```
You are provided with a list of artificial intelligence keywords and their
descriptions. Determine whether a subset of these keywords can be merged into a
single keyword based on their affiliation with the same subfield of artificial
intelligence.

- **Merge Criteria:**
- If at least two keywords belong to the same specific AI subfield, set `can_merge`
to `yes`; otherwise, set it to `no`.
- `keywords_to_merge` is a list of subsets of keywords that can be merged and does
not need to include all keywords.
- The `merged_keyword` must represent the most specific common concept and should
not be overly broad (e.g., not "artificial intelligence," "neural networks," or "
deep learning").
- The `merged_keyword` should be concise.
- The merged_description should focus solely on the merged_keyword itself, and does
not reference the keywords being merged.

**Input: keywords_and_descriptions**
{keywords_descriptions}

**Output:**
A structured JSON object with the following fields:
- If can_merge is "yes":
{{
  "can_merge": "yes",
  "merge_info": {{
    "keywords_to_merge": [list of keywords to merge],
    "merged_keyword": "new_keyword",
    "merged_description": "a brief introduction to the merged keyword" # Use the
    sentence pattern "{{merged_keyword}} is xxxx"
  }}
}}

- If can_merge is "no":
{{
  "can_merge": "no",
  "merge_info": null
}}

Begin your output here in json format:
```

