# OpenReview forum: "Agent Reviewers: Domain-specific Multimodal Agents with Shared Memory for Paper Review"
_ICML.cc/2025/Conference — ICML 2025 poster_

### Official Review · Reviewer_FTae · 2025-02-22

**Overall Recommendation:** 3

**Summary:**

This paper proposes a multi-agent system to simulate the review process of research papers, called Agent Reviewers. It is equipped with multi-agent interaction, shared memory pool, and multimodal agent. It empowers agent reviewers with observations not only on textual content but also on visual content. The experiments have shown an improvement compared with previous works. This work can benefit the research community, especially by providing more chances to polish their work for all researchers.

**Claims And Evidence:**

In the experiment section, the author presents abundant results to show the effectiveness of their works. I think they can verify the claims that the author has proposed.

**Essential References Not Discussed:**

There seem to be no essential references that are not discussed.

**Experimental Designs Or Analyses:**

The experimental designs are reasonable, with major experimental results and ablation studies. These experiments verify the claims proposed in Section 1, such as the capabilities of multimodality and SMP.

**Methods And Evaluation Criteria:**

The demonstration of the proposed method is clear. It designs multiple roles to simulate different reviewers during various stages. The review process is reasonable as well. The Share Memory Pool can further enhance the capability to provide responses of higher quality. The evaluation metrics are reasonable but should be further discussed in the main part of this paper.

**Other Comments Or Suggestions:**

Please see above.

**Other Strengths And Weaknesses:**

Please see above.

**Questions For Authors:**

- Are the results different among various conferences? For example, ICLR and NeurIPS.
- Are the results different among various years? For example, ICLR 2024 and ICLR 2023.

**Relation To Broader Scientific Literature:**

Yes, this paper can contribute to the research community.

**Theoretical Claims:**

There seem to be no theoretical claims.

---

> ### Author Rebuttal · Authors · 2025-03-31
>
> Thank you for the detailed comments.
>
> **Q1**: Suggestion of put the explanation of metrics in the main text of this paper.
>
> **A1**: Thank you for your suggestion. We agree that a clear explanation of metrics is crucial for understanding the paper. However, due to the page limit of the main text in ICML, we opted for a trade-off by providing qualitative descriptions in the main text and detailed explanations in Appendix B.1. We plan to provide a more detailed explanation of the metrics in the main text in a future version, which will be updated on arXiv after acceptance.
>
> **Q2**: Are the results different among various conferences? For example, ICLR and NeurIPS.
>
> **A2**:
> We have supplemented the results for ICLR 2024 and NeurIPS 2024. For each conference, 100 papers were randomly sampled for evaluation, and the default LLM GPT-4o-mini was used. The results are shown in the table below. Columns from Recall(S&W) to Jaccard(S&W) represent the analysis of strengths and weaknesses, while columns from F1(Dec.) to G-mean(Dec.) represent the analysis of decisions. "S&W" refers to strengths and weaknesses, and "Dec." stands for decisions. All metrics are better when larger.
> |Conference|Method|Recall(S&W)|F1(S&W)|MaxSim(S&W)|Jaccard(S&W)|F1(Dec.)|MCC(Dec.)|Bal.A(Dec.)|G-mean(Dec.)|
> |:-|:-|:-:|:-:|:-:|:-:|:-:|:-:|:-:|:-:|
> |ICLR 2024|Single Agent|0.347|0.416|0.442|0.270|0.469|0.219|0.609|0.593|
> |ICLR 2024|Agent Reviewers|0.409↑|0.453↑|0.476↑|0.307↑|0.605↑|0.385↑|0.705↑|0.705↑|
> |NeurIPS 2024|Single Agent|0.357|0.440|0.444|0.299|0.349|0.039|0.525|0.424|
> |NeurIPS 2024|Agent Reviewers|0.418↑|0.461↑|0.476↑|0.319↑|0.589↑|0.120↑|0.591↑|0.569↑|
>
> ICLR 2024 and NeurIPS 2024 produce similar outcomes, where Agent Reviewers outperform the Single Agent baseline, achieving an average relative improvement of  **23.5%**  on ICLR and  **44.7%**  on NeurIPS, thereby demonstrating consistency.
>
> **Q3**:Are the results different among various years? For example, ICLR 2024 and ICLR 2023.
>
> **A3**:
> We have supplemented the results for ICLR 2023, and ICLR 2024. For each conference, 100 papers were randomly sampled for evaluation, and the default LLM GPT-4o-mini was used. The cutoff year for the shared memory pool (SMP) was set to one year prior to the test data year, to avoid data leakage from SMP. The results are shown in the table below. Columns from Recall(S&W) to Jaccard(S&W) represent the analysis of strengths and weaknesses, while columns from F1(Dec.) to G-mean(Dec.) represent the analysis of decisions. "S&W" refers to strengths and weaknesses, and "Dec." stands for decisions. All metrics are better when larger.
> |Conference|Method|Recall(S&W)|F1(S&W)|MaxSim(S&W)|Jaccard(S&W)|F1(Dec.)|MCC(Dec.)|Bal.A(Dec.)|G-mean(Dec.)|
> |:-|:-|:-:|:-:|:-:|:-:|:-:|:-:|:-:|:-:|
> |ICLR 2024|Single Agent|0.347|0.416|0.442|0.270|0.469|0.219|0.609|0.593|
> |ICLR 2024|Agent Reviewers|0.409↑|0.453↑|0.476↑|0.307↑|0.605↑|0.385↑|0.705↑|0.705↑|
> |ICLR 2023|Single Agent|0.334|0.408|0.432|0.271|0.474|0.169|0.580|0.554|
> |ICLR 2023|Agent Rreviewers|0.390↑|0.439↑|0.467↑|0.298↑|0.538↑|0.139↓|0.570↓|0.570↑|
>
> Whether in 2023 or 2024, Agent Reviewers have shown better results than the single-agent baseline, demonstrating the consistency of the gains of our method. It should be noted that for ICLR 2023, due to the earlier cutoff year of the SMP, the number of papers in the SMP decreased from 28,372 to 21,156 (a 25% relative decrease), which may cause the performance decline in 2023.

---

### Official Review · Reviewer_pd6u · 2025-03-10

**Overall Recommendation:** 3

**Summary:**

This paper introduces a multi-agent system that enhances automated peer review. It mimics human review processes by employing domain-specific agents, a multimodal reviewer for visual analysis, and a shared memory pool (SMP) that retrieves past paper reviews for informed evaluation. The system also introduces *Reviews-STD*, a standardized dataset of paper reviews from ICLR and NeurIPS, formatted into strengths, weaknesses, and decisions. Tested on 300 ICLR 2024 papers, *Agent Reviewers* outperforms existing AI-based review systems.

**Claims And Evidence:**

The proposed system shows better results compared to baselines but the setting of the experiments (see below) and some questions (see below) need to be clarified to better review this paper.

**Essential References Not Discussed:**

The related work has discussed essential references

**Experimental Designs Or Analyses:**

I have thoroughly reviewed the experimental designs and identified several critical questions that need to be addressed:

First, there are concerns about the open-source model's performance metrics and how they compare to proprietary models. The paper would benefit from a more detailed analysis of these performance comparisons.

Second, the authors should clarify their plans for code and data availability. Making these resources open-source would greatly enhance reproducibility and allow the broader research community to build upon this work.

Third, there are potential data leakage concerns regarding the use of Gemini and Deepseek v3 models. The authors should address how they prevented any training data overlap between these models and their evaluation dataset.

**Methods And Evaluation Criteria:**

Please see questions below.

**Other Comments Or Suggestions:**

To strengthen the system's capabilities, I suggest implementing real-time online search functionality rather than relying exclusively on retrieval-augmented generation (RAG). This would allow the system to access the most current research and developments in the field.

Additionally, the system would benefit from expanding its memory pool to include papers from a broader range of conferences and journals beyond ICLR and NeurIPS. This expansion, combined with real-time search capabilities, would provide more comprehensive and up-to-date context for paper evaluations.

**Other Strengths And Weaknesses:**

The motivation and research question is important for this field and the proposed system seems promising but there are still quite a few questions need to be addressed for further review.

**Questions For Authors:**

I have several important questions for the authors that would help clarify key aspects of the system and potentially influence my evaluation:

First, I would like to understand the scope of background knowledge incorporated into the system. Which specific academic fields and domains are currently covered, and how comprehensive is this coverage?

Second, regarding data quality considerations, could you elaborate on how the presence of low-quality reviews in the training dataset affects the system's overall performance and reliability?

Third, I have concerns about the current architecture using a single LLM for multiple agent roles. This approach may limit the diversity of perspectives and capabilities. Have you considered evaluating the system's performance with different specialized LLMs assigned to different reviewer roles?

Fourth, the disparity between Strength and Weakness scores is notable. Could you provide insights into why the system appears to be less effective at identifying weaknesses compared to strengths in papers?

Fifth, what kind of LLMs are used in the baseline methods in Table 2? Line 307 indicates that these baselines follow their default configurations but it is unclear what LLM they are using.

Finally, the current analysis seems to focus primarily on titles, abstracts, and introductions (line 281). Could you explain this limitation and discuss any plans to extend the analysis to include the full content of papers?

**Relation To Broader Scientific Literature:**

This paper is related to the agent system for automatic paper reviews. The general idea is to develop automatic agent systems to review papers.

**Theoretical Claims:**

There is no theoretical claims in this paper.

---

> ### Author Rebuttal · Authors · 2025-03-31
>
> Thanks for your detailed and insightful comments.
>
> **Q1**:What kind of LLMs are used in the baseline methods in Table 2?
>
> **A1**:For the main experiment, all baselines and our Agent Reviewers use GPT-4o-mini for fair comparison (see Table 2 header). Other baseline settings like hyperparameters follow their defaults (Line 307, and detailed in Appendix B.2.). Thanks for your question; we'll clarify in the main text.
>
> **Q2**:Concerns about potential data leakage of Gemini and Deepseek v3.
>
> **A2**:We selected the latest ICLR 2024 data as the test set, taking measures to avoid data leakage. In main experiments, we used GPT-4o-mini (knowledge cutoff: Oct. 2023, no leakage). Gemini-exp-1206 (cutoff: Dec. 2023, no leakage) and Deepseek-V3 (cutoff: Jun. 2024, leakage not guaranteed despite efforts) were used only for cross-LLM generalization verification. The risk is explained in Appendix B.4.
>
> We contend Deepseek-V3 has no serious evaluation-affecting leakage due to:1)Overall performance is inferior to GPT-4o-mini(no leakage).2)When asked to detail papers by title, hallucinations are severe.3)Table 12 shows Agent Reviewers outperforms Single Agent with Deepseek-V3, indicating our method value.
>
> **Q3**:Background knowledge domains and coverage.
>
> **A3**:We currently focus on review in AI. Background knowledge comes from LLMs' inherent knowledge and the shared memory pool(SMP), which has 283728 AI papers and reviews from ICLR 2017-2023 and NeurIPS 2016-2023 on OpenReview, spanning nearly all AI fields. Notably, our training-free method can be applied to other fields by expanding SMP.
>
> **Q4**:Open-source model's performance.
>
> **A4**:Thanks for your suggestion. We've added comparisons with more open-source models (Deepseek-V3-0324, Qwen2.5-VL-72B-Instruct, InternVL-2.5-78B) in the [link](https://imgdrop.io/image/2iYyt).
>
> Overall, GPT-4o-mini performs best, Deepseek-V3 close behind. Largest open-source model approaches or surpasses some proprietary ones. Little difference exists between 2 Deepseek-V3 versions.
>
> **Q5**:Diversity concern & Proposal for different LLMs in different role experiments.
>
> **A5**:We agree on the importance of diversity in review, tried your suggestion with GPT-4o-mini, Gemini-exp-1206, and Deepseek-V3-0324 as LLMs for 3 domain-specific reviewers. Results at [link](https://imgdrop.io/image/2iI9B).
>
> We've seen no better results with a multi-LLM system. We analyze that our system has already enhanced review perspective diversity by endowing domain-specific reviewers with different domain knowledge, so multi-LLMs have limited room for improvement.
>
> **Q6**:Concerns about low-quality review impact in training data.
>
> **A6**:Our method is training-free. We initialized the SMP with ICLR 17-23 and NeurIPS 16-23 papers, and retrieved paper info(summaries, processed reviews) as domain-specific background knowledge for reviewers.
> We agree low-quality reviews can affect performance and mitigate it as follows: 1) Processed reviews in SMP, retaining AC decisions and LLM-aggregated  pros&cons to reduce individual low-quality review impact.2) Retrieved 5 domain-related papers for each reviewers as memory to ensure sufficient high-quality reviews.
>
> **Q7**:Why focus on title+abstr.+intro.& Plan to extend to full paper content.
>
> **A7**:We use only title+abstr.+intro. in our current method for two reasons: 1) The full text being about 12 times longer, using it in a multi-agent system with interactions incurs great overhead.2)They're crucial in peer review, even for humans. We agree full text can provide richer info and potentially better performance, but challenges like long context need to be overcome. To extend efficiently, we suggest adding an in-depth reading feature for reviewers. After reading first three parts, they can raise questions and retrieve relevant full-text parts for further analysis before generating comments.
>
> **Q8**:Insights into why system identifies paper weaknesses less effectively.
>
> **A8**:Thanks for noting this profound observation. We observed it in all tested  methods. We posit it's because reviewers' proposed strengths(str.) maybe similar, e.g., similar to claimed contributions, while weaknesses(wk.) vary greatly, making it harder for LLM-generated weaknesses to match humans'.
>
> We studied the similarity of Agent Reviewers' str. and wk. generated by GPT-4o-mini, Gemini, and Deepseek-V3 for the same article. On average, GPT-4o-mini vs. Gemini: str. 0.664/wk. 0.484; GPT-4o-mini vs. Deepseek-V3: str. 0.738/wk. 0.663. Figure 6 shows most matching str. in "Novelty" & "Research Implications", wk. dispersed.
>
> **Q9**:Suggestion about  real-time online search and memory pool expanding.
>
> **A9**:Thanks for your comments! We agree and will try these two approaches in future work to offer reviewers more comprehensive knowledge.
>
> **Q10**:Plans for code and data availability.
>
> **A10**:We'll open-source all code and data (include memory pool) upon acceptance for reproducibility and are preparing for it.

---

> > ### Comment · Reviewer_pd6u · 2025-04-03
> >
> > Thanks for the detailed response which has addressed most of my questions and I tend to increase my score by 1.

---

> > > ### Author Response · Authors · 2025-04-03
> > >
> > > Thanks again for the detailed comments and insightful suggestions!

---

### Official Review · Reviewer_GyWW · 2025-03-14

**Overall Recommendation:** 3

**Summary:**

This paper introduces Agent Reviewers, a multi-agent system designed to enhance peer review processes using Large Language Models (LLMs). The system comprises domain-specific agents with a shared memory pool (SMP) that enables them to incorporate historical knowledge and multimodal reviewers that assess visual elements of a paper. The authors construct the largest standardized dataset of paper reviews, Reviews-STD, and evaluate their system on ICLR 2024 submissions. The results indicate superior performance compared to existing AI-based review methods, demonstrating improved quality in review comments and acceptance predictions.

**Claims And Evidence:**

Claims

- Agent Reviewers generates more insightful and diverse reviews compared to existing AI-based review systems.

- The shared memory pool enhances domain-specific knowledge retrieval, leading to improved review accuracy.

- The multimodal reviewer contributes to better assessments by incorporating visual information.

Evidence
- Quantitative Evaluation: The system is benchmarked against AI Scientist, AgentReview, and LLM Review, demonstrating an 8.5% improvement in F1-score and a 10.5% increase in Jaccard index for strengths-weaknesses alignment.
- Decision Analysis: The system achieves a 35.7% improvement in decision F1-score and a 78.9% increase in MCC over existing methods.
- Ablation Studies: Removing the multi-agent discussion or shared memory pool leads to a decline in review accuracy, confirming their effectiveness.
- Case Studies: Examples show how the system provides richer, context-aware critiques by referencing prior research.

**Essential References Not Discussed:**

The related work is comprehensive

**Experimental Designs Or Analyses:**

- Benchmarking Against Existing Systems: Compared to AI Scientist, AgentReview, and LLM Review on ICLR 2024 papers.
- Ablation Studies: Tested the impact of removing multi-agent collaboration, shared memory, and multimodal components.
- Impact of Shared Memory Initialization: Evaluated how the cutoff year of included papers affects review quality.

**Methods And Evaluation Criteria:**

System Architecture:
- Multi-Agent Interaction (MI): Different agents collaborate in reviewing, discussion, and final decision-making.
- Shared Memory Pool (SMP): Enables domain-specific knowledge retrieval from prior papers.
- Multimodal Reviewer (MA): Evaluates visual aspects like figures and layout.

**Other Comments Or Suggestions:**

How is the NeurIPS dataset obtained, given that only the reviews for accepted papers are released?

**Other Strengths And Weaknesses:**

See above

**Questions For Authors:**

See above

**Relation To Broader Scientific Literature:**

This work builds on / is relevant to AI-assisted peer review, Multi-agent systems for research automation (AI Scientist, CycleReviewer). Multimodal AI models incorporating textual and visual data.

**Theoretical Claims:**

This is an application paper.

---

> ### Author Rebuttal · Authors · 2025-03-31
>
> Thank you for the detailed comments.
>
> **Q1**: How is the NeurIPS dataset obtained, given that only the reviews for accepted papers are released?
>
> **A1**: All our data(papers and reviews) stems from the public data on OpenReview, including the NeurIPS dataset. As you noted, 96% of the NeurIPS papers (2016 - 2024) released on OpenReview were accepted. We describe this statistic in Appendix A. Due to the severe imbalance in the NeurIPS data, we do not use it as test data. Instead, we use ICLR 2024 for evaluation.
>
> We use NeurIPS 2016 - 2023 and ICLR 2017 - 2023 data to initialize the shared memory pool, providing domain-specific reviewers with sufficient  background knowledge for review. Although nearly all NeurIPS dataset papers are accepted, the ICLR dataset has an acceptance rate of about 37%. Consequently, papers in the shared memory pool are relatively evenly balanced in acceptance and rejection.

---

### Decision · Program_Chairs · 2025-05-01

**Decision:**

Accept (poster)

**Comment:**

This paper proposes Agent Reviewers, a multimodal multi‑agent framework that aims to automates scholarly peer review. A large corpus of historical reviews populates a shared memory pool that agents query while collaboratively critiquing both the textual and visual content of a manuscript. The system outperforms prior AI review tools on strengths/weakness alignment and decision prediction; ablation studies confirm the importance of multi‑agent discussion, the memory pool, and the multimodal agent.

The reviewers agree the paper tackles an important practical problem. The constructed peer-review dataset can be useful to the research community. The proposed method is clear. The experiments are comprehensive: comparisons with three baselines (AI Scientist, AgentReview, and LLM Review), ablations, cross‑LLM generalization, and new results added during rebuttal for venues of NeurIPS 2024 and ICLR 2023/24.

The original submission lacked clarity regarding evaluation metrics, dataset sourcing, data leakage safeguards, and the choice of baseline LLM. It seems the rebuttal addressed most of these points satisfactorily.
One major issue is the analysis currently focuses only on titles, abstracts, and introductions sections of the paper, which does not fully reflect the real-world peer-review process. The authors explain this limitation is due to challenges in handling long contexts, and defer the full-paper analysis and real-time retrieval to future work. The system's weaker coverage of weaknesses versus strengths remains an open issue – the authors should discuss this in the future revisions.